# A composable machine-learning approach for steady-state simulations on high-resolution grids

**Rishikesh Ranade**
Office of CTO
Ansys Inc.
Canonsburg, PA 15317
`rishikesh.ranade@ansys.com`

**Chris Hill**
Fluids Business Unit
Ansys Inc.
Lebanon, NH, 03766
`chris.hill@ansys.com`

**Lalit Ghule**
Office of CTO
Ansys Inc.
Canonsburg, PA, 15317
`lalit.ghule@ansys.com`

**Jay Pathak**
Office of CTO
Ansys Inc.
San Jose, CA, 95134
`jay.pathak@ansys.com`

## Abstract

In this paper we show that our Machine Learning (ML) approach, CoMLSim (**Co**mposable **M**achine **L**earning **Sim**ulator), can simulate PDEs on highly-resolved grids with higher accuracy and generalization to out-of-distribution source terms and geometries than traditional ML baselines. Our unique approach combines key principles of traditional PDE solvers with local-learning and low-dimensional manifold techniques to iteratively simulate PDEs on large computational domains. The proposed approach is validated on more than 5 steady-state PDEs across different PDE conditions on highly-resolved grids and comparisons are made with the commercial solver, Ansys Fluent as well as 4 other state-of-the-art ML methods. The numerical experiments show that our approach outperforms ML baselines in terms of 1) accuracy across quantitative metrics and 2) generalization to out-of-distribution conditions as well as domain sizes. Additionally, we provide results for a large number of ablations experiments conducted to highlight components of our approach that strongly influence the results. We conclude that our local-learning and iterative-inferencing approach reduces the challenge of generalization that most ML models face.

## 1 Introduction

Engineering simulations utilize solutions of partial differential equations (PDEs) to model various complex physical processes like weather prediction, combustion in a car engine, thermal cooling on electronic chips etc., where the complexity in physics is driven by several factors such as Reynolds number, heat source, geometry, etc. Traditional PDE solvers use discretization techniques to approximate PDEs on discrete computational domains and combine them with linear and non-linear equation solvers to compute PDE solutions. The PDE solutions are dependent on several conditions imposed on the PDE, such as geometry and boundary conditions of the computational domain as well as source terms such as heat generation, buoyancy etc. Moreover, many applications in engineering require highly-resolved computational domains to accurately approximate the PDEs solutions as well as to capture the intricacies of conditions such as geometry and source terms imposed on the PDEs. As a result, traditional PDE solvers, although accurate and generalizable across various PDE conditions, can be computationally slow, especially in the presence of complicated physics and

36th Conference on Neural Information Processing Systems (NeurIPS 2022).

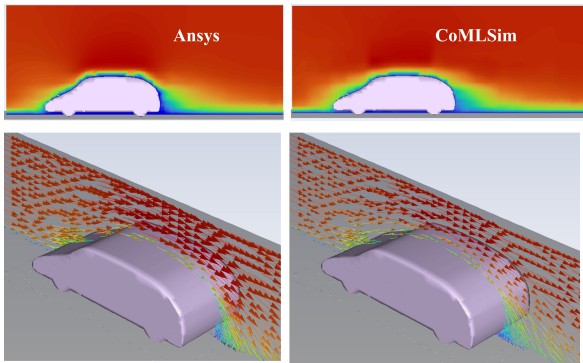

Figure 1: CoMLSim performance vs Ansys Fluent for a high Reynolds number, steady-state turbulent flow over a car (never seen during training) at 90 mph. The vectors and contours predictions match reasonably with MAE of 0.0145. High flow gradients that affect the wake pattern are resolved accurately and comparable to Fluent.

large computational domains. In our work, we specifically aim to increase the speed of engineering simulations using ML techniques on complex PDE conditions used for enterprise simulations.

The idea of using Machine Learning (ML) with PDEs is not unique and has been explored for several decades [1, 2]. ML approaches are computationally fast, but they fall short in terms of accuracy, generalization to a wide range and out-of-distribution PDE conditions and in their ability to scale to highly-resolved computational grids, when compared to traditional PDE solvers. A few shortcomings of the current methods are outlined below and subsequently are motivations for this work.

1. ML approaches use static inferencing to predict PDE solutions as a function of the PDE conditions. In many cases, solutions and conditions have high-dimensional and sparse representations, which are challenging to generalize with such black-boxed static inferencing.

2. Most engineering applications require highly-resolved computational grids to capture detailed solution features, such as hot spots on an electronic chip surface. This adversely impacts training in terms of GPU memory, computational time and data requirements.

3. Most ML approaches fail to use powerful information from traditional PDE solvers related to solver methods, numerical discretization etc.

In this work, we introduce a novel ML approach, **Co**mposable **M**achine **L**earning **Sim**ulator (CoMLSim) to work with high-resolution grids without compromising accuracy. To achieve that, our method decomposes highly-resolved computational grids into smaller local subdomains with Cartesian grids. PDE solutions and conditions on each subdomain are represented by lower-dimensional latent vectors determined using autoencoders. The latent vectors of solutions corresponding to conditions are predicted in each subdomain and stitched together to maintain local consistency, analogous to flux conservation in traditional solvers. In this paper, we describe our method (shown in fig. 2) for building CoMLSim for different PDEs, showcase different experiments to compare accuracy and generalizability with different ML methods and provide an extensive ablation study to understand the impact of CoMLSim's different components on the results.

**Significant contributions of this work:**

1. The CoMLSim approach combines traditional PDE solver strategies such as domain discretization, flux conservation, solution methodologies etc. with ML techniques to accurately model numerical simulations on high-resolution grids.

2. Our approach operates on local subdomains and solves PDEs in a low-dimensional space. This enables generalizing to out-of-distribution PDE conditions and scaling to bigger domains with large mesh sizes.

3. The iterative inferencing algorithm is self-supervised and allows for coupling with traditional PDE solvers.

**Related Works:** The use of ML for solving PDEs has gained tremendous traction in the past few years. Much of the research has focused on improving neural network architectures and optimization

techniques to enable generalizable and accurate learning of PDE solutions. More recently, there has been a lot of focus on learning PDE operators with neural networks (NNs) [3, 4, 5, 6, 7, 8, 9]. The neural operators are trained on high-fidelity solutions generated by traditional PDE solvers on a computational grid of specific resolution and do not require any knowledge of the PDE. However, accuracy and out-of-distribution generalizability deteriorates as the resolution of computational grid increases (for example, $2048^2$ in 2-D or $256^3$ in 3-D). Furthermore, these methods are limited by an upper cap on GPU memory.

A different research direction focuses on training neural networks with physics constrained optimization [10, 11]. These method use Automatic Differentiation (AD) [12] to compute PDE derivatives. The physics-based approaches have been extended to solve complicated PDEs representing complex physics [13, 14, 15, 16, 17, 18, 19, 20, 21, 22, 23, 24, 25, 26, 27, 28, 29, 30]. More recently, alternate approaches that use discretization techniques using higher order derivatives and specialize numerical schemes to compute derivatives have shown to provide better regularization for faster convergence [31, 32, 33, 34]. However, the use of optimization techniques to solve PDEs, although accurate, has proved to be extremely slow as compared to traditional solvers and hence, non-scalable to high resolution meshes.

Domain decomposition techniques have been successfully used in the context of ML applied to PDEs [35]. These techniques enable local learning from smaller restricted domains and have proved to accelerate learning of neural networks. In [36, 35], the authors proposed a hybrid method combining ML with domain decomposition to reduce the computational cost of finite element solvers. Similarly, domain decomposition has been extensively used in the context of physics informed neural networks to reduce training costs by enabling distributed training on multiple GPUs [37, 38]. [39] and [40] learn on localized domains but infer on larger computational domains using a stitching process. [41] and [42] learn coefficients of numerical discretization schemes from high fidelity data, which is sub-sampled on coarse grids. [43] learn surrogate models for smaller components to allow for cheaper simulations. [44] learn local prolongation operators from discretization matrices, to improve the rate of convergence of the multigrid linear solvers. Alternatively, other methods such as [45, 46, 47] make us of graph-based models to improve generalization by learning on local elements. In this work, we use a Cartesian grid to discretize each subdomain. However, GNNs [48, 49, 50, 51, 52] as well as FEM use other discretizations such as triangle or polygonal meshes. In future, these ideas can be used to handle unstructured meshes in subdomains.Additionally, other ML methods improve the learning of ML models by compressing PDE solutions on to lower-dimensional manifolds. This has shown to improve accuracy and generalization capability of neural networks [53, 54, 55, 56, 57, 58]. Although, here we propose an iterative inferencing approach in the context of solving PDEs, it is inspired from the implicit deep learning approach used in a variety of machine learning tasks [59, 60, 61, 62]. In this work, domain decomposition is combined with latent space learning is employed to accurately represent solutions on local subdomains and allow scalability to bigger computational domains.

## 2 CoMLSim approach

### 2.1 Similarities with traditional PDE solvers

Consider a set of coupled steady-state PDEs with $n$ solution variables ($n = 2$ for example). The coupled PDEs are defined as follows:

$$L_1(u,v) - F_1 = 0; L_2(u,v) - F_2 = 0 \qquad (1)$$

$u(x,y,z)$ and $v(x,y,z)$ are defined on a computational domain $\Omega$ with boundary conditions specified on the boundary of the computational domain, $\Omega_b$. Here, $L_1$, $L_2$ denote PDE operators, $F_1$, $F_2$

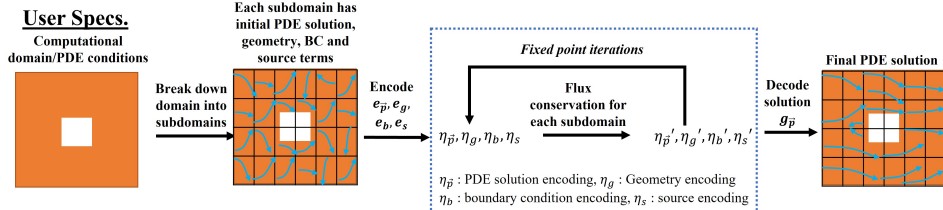

Figure 2: CoMLSim solution algorithm

represent PDE source terms and $u = u_b, v = v_b$ for $\Omega = \Omega_b$. The PDE operators can vary for different PDEs. For example, in a non-linear PDE such as the unsteady, incompressible Navier-Stokes equation the operator, $L = \vec{a}.\vec{\nabla} - \vec{\nabla}.\vec{\nabla}$ Traditional FVM/FDM based PDE solvers solve PDEs in Eq. 1 by computing solutions variables, $u, v$, and their linear/non-linear derivatives on a discretized computational domain. Iterative solution algorithms are used to conserve fluxes between neighboring computational elements and determine consistent PDE solutions over the entire domain at convergence. The CoMLSim approach is designed to perform similar operations but at the level of subdomains ($n^3$ elements) using ML techniques. Additional details are provided in supplementary materials sections A.4.

## 2.2 Solution algorithm for steady-state PDEs

The solution algorithm of CoMLSim approach is shown in Fig. 2 and described in detail in Alg. 1. Similar to traditional solvers, the CoMLSim approach discretizes the computational domain into smaller subdomains contain $n^p$ computational elements, where $p$ refers to spatial dimensionality and $n$ is predetermined. Each subdomain has a constant physical size and represents both PDE solutions and conditions. For example, a subdomain cutting across the cylinder in Fig. 2 represents the geometry and boundary conditions of the part-cylinder as well as the corresponding solution. In a steady-state problem, the solutions on the domain are initialized either with uniform solutions or they are generated from coarse-grid PDE solvers. The initial solution $\vec{p} = (u(\vec{x}), v(\vec{x}))$ is shown with randomly oriented flow vectors in Fig. 2. Pretrained encoders $e_{\vec{p}}, e_g, e_b, e_s$ are used to encode the initial solutions as well as user-specified PDE conditions into lower-dimensional latent vectors, $\eta_{\vec{p}}$ corresponding to PDE solutions and $\eta_g, \eta_b, \eta_s$ corresponding to geometry, boundary conditions and source terms, respectively. Solution and condition latent vectors on groups of neighboring subdomains are concatenated together evaluated using a pre-trained flux conservation autoencoder $(\Theta)$ to get a new set of solution latent vectors $(\eta'_{\vec{p}})$ on each subdomain, which are more locally consistent than the original vectors. The new solution latent vectors are iteratively passed through $\Theta$ to improve their local consistency and the iteration is stopped when the $L_2$ norm of change in solution latent vectors meets a specified tolerance, otherwise the iteration continues with the updated latent vectors. The latent vectors of the PDE conditions are not updated and help in steering the solution latent vectors to an equilibrium state that is decoded to PDE solutions using pretrained decoders $(g_{\vec{p}})$ on the computational domain. The converged solution in Fig. 2 is represented with flow vectors that are locally consistent with neighboring subdomains. The iterative procedure used in the CoMLSim approach can be implemented using several linear or non-linear equation solvers, such as Fixed point iterations [59], Gauss Seidel, Newton's method etc., that are used in commercial PDE solvers. Out of those we have explored Point Jacobi and Gauss Seidel, which are described below. The use of autoencoders for compressed solution and condition representation in tandem with iterative inferencing solution algorithm is inspired from [58, 31] but the main difference in this work is that the solution procedure is carried out on local subdomains as opposed to entire computational domains to align with principles used in traditional PDE solvers. Supplementary material section G explains our algorithm in more details using an example.

---

**Algorithm 1:** Solution methodology of CoMLSim approach

---

1   Domain Decomposition: Computational domain $\Omega \to$ Subdomains $\Omega_c$
2   Initialize solution on all $\Omega_c$: $\vec{p}(\vec{x}) = 0.0$ for all $\vec{x} \in \Omega_c$
3   Encode solutions on $\Omega_c$: $\eta_{\vec{p}} = e_u(\vec{p}(\Omega_c))$
4   Encode conditions on $\Omega_c$:
5   $\eta_g = e_g(g(\Omega_c)), \eta_b = e_b(b(\Omega_c)), \eta_s = e_s(s(\Omega_c))$
6   Set convergence tolerance: $\epsilon_t = 1e^{-8}$
7   **while** $\epsilon > \epsilon_t$ **do**
8     **for** $\Omega_c \in \Omega$ **do**
9       Gather neighbors of $\Omega_c$: $\Omega_{nb} = [\Omega_c, \Omega_{left}, \Omega_{right}, ...]$
10       $\eta'_{\vec{p}} = \Theta(\eta_{\vec{p}}^{nb}, \eta_b^{nb}, \eta_g^{nb}, \eta_s^{nb})$
11     Compute $L_2$ norm: $\epsilon = ||\eta_{\vec{p}} - \eta'_{\vec{p}}||_2^2$
12     Update: $\eta_{\vec{p}} \leftarrow \eta'_{\vec{p}}$ for all $\Omega_c \in \Omega$
13   Decode PDE solution on all $\Omega_c$: $\vec{p} = g_{\vec{p}}(\eta_{\vec{p}\Omega_c})$

---

**Why flux conservation networks work?** The flux conservation network ($\Theta$) is simply an autoencoder which takes encoding of solutions and conditions on a group of neighboring subdomains as both inputs and outputs. Similar to traditional solvers that use approximations to represent relationships between neighboring solutions on elements, the flux conservation autoencoder does the same for encoded subdomain solutions and corresponding conditions. The training of this network is carried out on converged, locally consistent solutions generated for a specific PDE for arbitrary conditions. Since this network has only learnt locally consistent solutions, if one were to initialize a group of neighboring subdomains with random noise and iteratively pass it through this network the output of such a procedure would be an equilibrium solution corresponding to some locally consistent PDE solution. However, the correct PDE solution it converges to depends on the fixed condition specified in this procedure. Additional results and details are provided in the supplementary materials sections A.1, A.4 and C.1.5.

**Stability of flux conservation autoencoder:** The iterative inferencing approach proposed in this work is similar to traditional approaches used in solving system of linear equations. Our algorithm uses fixed point iterations to solve Eq. 2

$$\eta'_{\vec{p}} = \Theta(\eta^{nb}_{\vec{p}}, \eta^{nb}_b, \eta^{nb}_g, \eta^{nb}_s) \tag{2}$$

where, $\Theta$ corresponds to the weights of flux conservation network, $\eta_{()}$ refers to the latent vectors and $nb$ refers to neighboring subdomains defined below in Eq. 3.

$$\eta^{nb}_{\vec{p}} = \left[ \eta^s_{\vec{p}}, \eta^{left}_{\vec{p}}, \eta^{right}_{\vec{p}}, \eta^{top}_{\vec{p}}, \eta^{bottom}_{\vec{p}} \right] \tag{3}$$

Similar to linear systems, the stability of our iterative inferencing approach is governed by the condition number [63] defined in Eq. 4

$$|| \frac{\partial \Theta(\eta_{\vec{p}}, \eta_b, \eta_g, \eta_s)}{\partial \eta_{\vec{p}}} || < 1 \tag{4}$$

**Solution algorithms: Point Jacobi vs Gauss Seidel:** Since the algorithm loops over all the subdomains in a specified order, while updating the solution encoding of a subdomain $c$, there are subdomains in the neighborhood that already have an updated solution encoding. The Gauss Seidel method uses these updated solution encodings ($\eta'_{\vec{p}}$) on neighboring subdomains to update the solution encoding on subdomain, $c$. On the other hand, Point Jacobi method does not make use of this and hence can be easily vectorized for significantly faster computation. The implementation has a remarkable similarity with how traditional solvers use these methods [64] and leaves the door open to use other non-linear optimizers with physics-based constraints.

## 2.3 Neural network components in CoMLSim

Our algorithm employs two types of autoencoders, CNN autoencoders to establish a lower-dimensional representation of PDE solutions and conditions from local subdomains grids and FCNN Autoencoders for flux conservation, where the goal is to learn a reduced representation of solution and condition latent vectors in a local neighborhood. Let us consider an example of solving the Laplace equation, $\nabla^2 u = 0$ in 2D for arbitrarily shaped computational domains, $\Omega_g$. In this example, the CoMLSim algorithm will require 3 autoencoders, described in Eq. 5, to learn lower-dimensional representations on local subdomains.

$$\left. \begin{array}{c} u' = S(u, \eta_u) \\ g' = G(g, \eta_g) \\ \eta'_{nb} = \Theta(\eta_{nb}, \zeta) \end{array} \right\} \tag{5}$$

where, $u, \eta_u, S$ refers to the solution, its latent vector and the weights of the autoencoder, respectively, $g, \eta_g, G$ refers to a representation of geometry, such as Signed Distance Fields (SDF) [65], its latent vector and the weights of the autoencoder, $\Theta$ refers to the weights of flux conservation autoencoder and $\eta_{nb}$ represents a set of concatenated latent vectors, $\eta_u$ and $\eta_g$ on a group of neighboring subdomains. All the autoencoders are trained with samples of PDE solutions generated for the same use case. In the case of coupled PDEs, a single autoencoder is trained for all solution variables. Additional details related to the autoencoder networks are provided in the supplementary materials section A.

**Why Autoencoders?:** Solutions to classical PDEs such as the Laplace equation can be represented by homogeneous solutions as follows:

$$\phi(x, y) = a_0 + a_1 x + a_2 y + a_3(x^2 - y^2) + a_4(2xy) + ... \tag{6}$$

where, $\vec{A} = a_0, a_1, a_2, ..., a_n$ are constant coefficients that can be used to reconstruct the PDE solution on any local subdomain. $\vec{A}$ can be considered as a compressed encoding of the Laplace solutions. Since, it is not possible to explicitly derive such compressed encodings for other high dimensional and non-linear PDEs, the CoMLSim approach relies on autoencoders to compute them. It is known that non-linear autoencoders with good compression ratios can learn powerful non-linear generalizations [66, 67, 68]. Autoencoders also have great denoising abilities, which improve robustness and stability, when used in iterative settings [58]. In their paper, Park and Lee [69] demonstrate that the latent manifold established by a trained autoencoder is stable for varying intensities of Gaussian noise. This quality of autoencoders is useful for the convergence of the iterative inferencing approach.

## 3 Experiments

In this section as well as in the supplementary materials section C, we consider a number of use cases with varying degrees of difficulty resulting from the PDE formulation as well as source terms, geometry and boundary conditions. The PDEs have applications in fluid mechanics, structural mechanics and semiconductor simulations.

**Details of experiments:** Here we provide some details about the experiments. Additional details and experiments on Laplace and Darcy equation may be found in the supplementary materials section D.

1. **2D Poisson equation:** The Poisson's equation, shown in Eq. 7, is very popular in engineering simulations, for example, chip temperature prediction, pressure equation in fluids etc.

$$\nabla^2 u = f \tag{7}$$

where, $u$ is the solution variable and $f$ is the source term. This PDE is solved on a 1024x1024 grid to resolve the high-frequency features of the source term. The source, $f$, is sampled from a Gaussian mixture model shown in Eq. 8.

$$\sum_{j=0}^{1024}\sum_{j=0}^{1024} f_{i,j} = \sum_{j=0}^{1024}\sum_{j=0}^{1024}\sum_{k=0}^{30} A_k \exp\left(-\left(\frac{x - \mu_{x,k}}{\sigma_{x,k}}\right)^2 - \left(\frac{y - \mu_{y,k}}{\sigma_{y,k}}\right)^2\right) \tag{8}$$

where, $x, y$ correspond to the grid coordinates. $A_k$ randomly assumes either 0 or 1 to vary the number of active Gaussians in the model. $\mu_x, \mu_y$ and $\sigma_x, \sigma_y$ are the mean and standard deviations of Gaussians in $x$ and $y$ directions, respectively. The means and standard deviations vary randomly between 0 to 1 and 0.001 to 0.01, respectively. The smaller magnitude of standard deviation results in hot spots that require highly-resolved grids. In this case, 256 training and 100 testing solutions are generated using Ansys Fluent.

2. **2D non-linear coupled Poisson equation:** The coupled non-linear Poisson's equation is shown below in Eq. 9. $u, v$ are the solution variables and $f$ is the source term, similar to the description in Eq. 8. This PDE has applications in reactive flow simulations.

$$\nabla^2 u = f - u^2; \qquad \nabla^2 v = \frac{1}{u^2 + \epsilon} - v^2 \tag{9}$$

The data generation is similar to Experiment 2 with a slight difference that the source term is less stiff with a standard deviation that varies between 0.005 to 0.05.

3. **3D Reynolds-Averaged Navier-Stokes external flow:** This use case consists of a 3-D channel flow with resolution 304x64x64 at high Reynolds number over arbitrarily shaped objects. The corresponding PDEs are presented in supplementary materials section C.3 or may be referred from [70]. The characteristics of flow generated on the downstream has important applications in the design of automobiles and airplanes. In this case, we use 5 primitive 3-D geometries namely, cylinder, cuboid, trapezoid, airfoil wing and wedge and their random combinations and rotational augmentations to create 150 training geometries and 50 testing geometries, which are solved with Ansys Fluent to generate the data. Out-of-distribution testing is carried out on simplistic automobile geometries as shown in Figure 1.

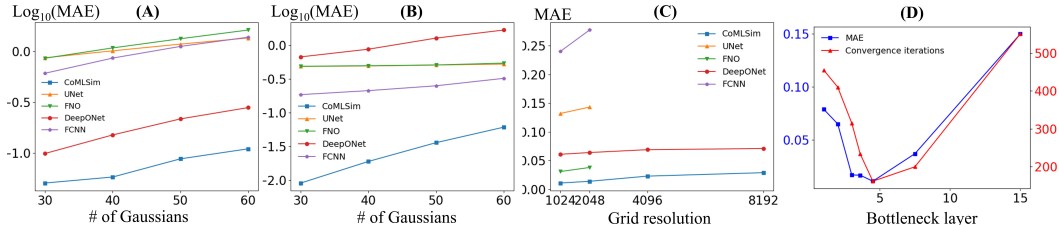

Figure 3: (A): Error vs No. of Gaussians for linear Poisson's; (B): Error vs No. of Gaussians for non-linear Poisson's; (C): Error vs grid resolution for linear Poisson's; (D): MAE and convergence iterations vs flux conservation bottleneck size

4. **3D chip cooling with Natural convection:** In this case, we extend the complexity of Experiments 1, 2 and 3 to a industrial 3-D case of chip cooling with natural convection. In this problem, the computational grid ($128^3$ resolution) consists of a chip that is subjected to the powermap specified by Eq. 8. The heating of the chip results in generation of velocity in the fluid and at steady-state, a balance is achieved. We use Ansys Fluent to generate 300 training and 100 testing solutions for velocity, pressure, temperature for arbitrary sources.

# 4 Results and Ablation studies

In this section, we provide a variety of results for the experiments outlined in Section 3. Additionally, we conduct thorough ablation studies to understand the different components of the CoMLSim approach in more detail. We use the 2-D Linear Poisson's equation experiment for these studies unless otherwise specified. Details regarding the CoMLSim set up for each experiment, baseline network architectures as well as additional results corresponding to contour and line plots and comparisons across other metrics are provided in supplementary materials section C.

## 4.1 Comparison with Ansys Fluent and other ML baselines

The CoMLSim is compared with other ML baselines namely, UNet [71], FNO [8] and DeepONet [7] for the experiments outlined in Section 3. We compare the mean absolute error with respect to Ansys Fluent and averaged over the unseen testing cases and all solution variables. In the case of 3-D chip cooling we report the $L_\infty$ norm of temperature as this metric is more suited to this industrial application. It may be observed from Table 1 that the CoMLSim approach performs better than other ML baselines. All the ML methods perform better for the non-linear Poisson's case as compared to the linear Poisson's because the source term is less stiff but the CoMLSim approach does a better job in modeling the non-linearity arising due solution coupling. It must be noted that CoMLSim as well as the baselines are trained with reasonably training samples. The baseline performance can have different outcomes with increasing the size of the training data.

Table 1: Comparison with baselines

| Experiment | Metric | CoMLSim | UNet | FNO | DeepONet | FCNN |
|---|---|---|---|---|---|---|
| 2-D Linear Poisson's | $L_1$ | 0.011 | 0.132 | 0.031 | 0.061 | 0.267 |
| 2-D non linear Poisson's | $L_1$ | 0.0053 | 0.0877 | 0.0278 | 0.527 | 0.172 |
| 3-D NS external flow | $L_1$ | 0.012 | 0.0625 | 0.038 | 0.81 | 0.125 |
| 3-D chip cooling | $L_\infty$ | 15.2 | 95.21 | 60.836 | 45.27 | 192.7 |

## 4.2 Assessment of generalizability

In this section, we assess the generalizability of our method specifically with respect to out-of-distribution PDE conditions and scaling to higher resolution grids. The results are compared with Ansys Fluent as well as other ML baselines discussed in section 4.1.

**Out-of-distribution source term:** The source term for linear and non-linear Poisson equations are sampled from a Gaussian mixture model described in Eq. 8, where the number of Gaussian is randomly chosen between 0 and 30. In this experiment, we evaluate

the generalizability of our approach for source terms with exactly 30, 40, 50 and 60 number of Gaussians, corresponding to out-of-distribution for the training data distributions. For each case, the results averaged over 10 testing samples are compared with Ansys Fluent.

It may be observed from 3A and B, that the accuracy of all ML approaches decreases as you move further away from the source term distribution. However, the CoMLSim approach performs significantly better than other ML baselines and the accuracy is reasonable even for the case with 60 Gaussian mixture model, which is substantially different from the training distribution.

**Increasing mesh size by increasing domain size:** Next, we evaluate the performance of CoMLSim for bigger domains with larger grid size and compare it with other ML baselines. In this experiment, we compute solutions for the 100 source term conditions in the test set but on 4 different domains with mesh sizes, $1024^2$, $2048^2$, $4096^2$, $8192^2$, respectively. The mean and standard deviation of Gaussians in the testing set are proportionally increased with the mesh size to ensure that the problem definition does not change. It is important to note that none of the networks are retrained for larger sized grids.

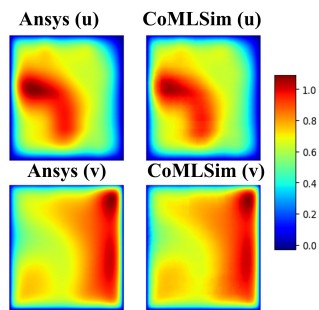

Figure 4: Example of out-of-distribution source (60 Gaussians) comparison for nonlinear Poisson's

It maybe observed from table 3 that the CoMLSim continues to scale to larger mesh sizes with similar accuracy as the original mesh size. The slight drop in accuracy due to increase in mesh size is attributed to accumulation of error caused by the flux conservation networks resulting from the increase in number of subdomains. Other ML baselines such as UNet and FNO scale up to a grid size of $2048^2$ but cannot evaluate beyond that due to GPU memory constraints. DeepONet has a point-wise inference and can also scale to larger-sized grids but its accuracy is lower than CoMLSim.

**Out-of-distribution geometries:** Next, we evaluate the performance of CoMLSim on 2 unseen car models presented at different 3-D angles of rotation to the external flow. It should be noted that these geometries are more complicated than the primitive objects considered for training our approach 3. The mean absolute errors with respect to Ansys Fluent across all solution variables are 0.0145, 0.06924, 0.04175, 0.91 and 0.13725 for CoMLSim, UNet, FNO, DeepONet and FCNN, respectively. Additional results and analysis are provided in supplementary materials C.3.3 and C.3.4.

### 4.3 Analysis of Subdomain size

In this experiment, we analyze the effect of subdomain resolution on accuracy and computational cost of the CoMLSim approach. We train 3 instances at resolutions of $32^2$, $64^2$ and $128^2$. The compression ratio in autoencoders is kept the same for the different subdomain resolutions. The mean absolute errors on testing set are 0.015, 0.011 and 0.029, respectively. Additionally, the iterations required to convergence are 420, 162 and 89, respectively. The convergence history is shown in Fig. 5A. The $128^2$ subdomain resolution has a lower accuracy because it is challenging to train accurate autoencoders as bigger subdomains capture a large amount of information. On the other hand, the computational cost is the lowest for the $128^2$ resolution because the number of subdomains in the entire computational domain are significantly less and hence, the solution algorithm converges faster.

### 4.4 Flux conservation bottleneck layer size

The flux conservation autoencoder as it is the primary workhorse of the CoMLSim solution algorithm. As a result, we evaluate the effect of the bottleneck layer size on the accuracy and computational speed. Each instance of CoMLSim with different bottleneck size is tested on 100 unseen test cases. We verify that all models satisfy the stability criterion specified in Eq. 3. It may be observed from Figure 3D that as the compression ratio of the flux conservation autoencoder decreases, it begins to overfit and the testing error as well as the number of convergence iterations and computational time significantly increase. On the other hand, if the compression ratio is too large the testing error increases because the autoencoders underfit. In alignment with the collective intuition about autoencoders, there exists an optimum bottleneck size compression ratio where the best testing error is obtained for small computational times.

## 4.5 Robustness and stability

A long standing challenge in the field of numerical simulation is to guarantee the stability and convergence of non-linear PDE solvers. However, we believe that the denoising capability of autoencoders [72, 66, 73, 74, 75] used in our iterative solution algorithm presents a unique benefit, irrespective of the choice of initial conditions. Here, we empirically demonstrate the robustness and stability of our approach. In scenario $A$, we randomly sample 25 initial solutions from a uniform distribution and in scenario $B$ we sample from 6 different distributions, namely Uniform, Gumbel, Laplace, Gamma, Normal, Logistic etc. The mean convergence trajectory is plotted in Fig. 5C and 5D show that the log of $L_2$ norm of the convergence error falls below an acceptable tolerance for all cases and demonstrates the stability of our approach.

## 4.6 CoMLSim
## solution methods: Point Jacobi vs Gauss Seidel

Next, we compare the performance of the Point Jacobi and Gauss Seidel implementations in the CoMLSim approach across 3 metrics, mean absolute error, computational cost and iterations to converge. Both methods have a similar prediction accuracy with a mean absolute error of $0.013$ and $0.017$, respectively but the Point Jacobi method is computationally cheaper requiring $4.3$ seconds to converge as compared to $72$ seconds of Gauss Seidel. It may be observed from Fig. 5E that the Gauss Seidel method takes significantly fewer convergence iterations, but the Point Jacobi method is faster because it can be efficiently vectorized on a GPU.

## 4.7 Coupling
## with traditional PDE solvers and super-resolution

In this experiment, we analyze the potential of coupling the CoMLSim approach with a traditional PDE solver such as Ansys Fluent and subsequently demonstrate how our approach inherently possesses the capability to perform super-resolution. We mainly carry out 5 experiments, where we start from a coarse grid solution for the 100 test samples of linear Poisson's equation solved by Ansys Fluent on a $64x64$ grid ($256x$ smaller than the fine resolution), add random noise to the coarse initial solution with varying amplitudes ($10\%$, $25\%$ and $50\%$), and use that as an initial solution in the CoMLSim approach. The convergence trajectory in Fig. 5B shows that the case with coarse grid initialization converges the fastest, followed by initialization with noise and zero initialization. All solutions converge to the same accuracy.

## 4.8 Computational speed

We observe that the CoMLSim approach is about $40-50x$ faster as compared to commercial steady-state PDE solvers such as Ansys Fluent for the same mesh resolution and physical domain size in all the experiments presented in this work. In comparison to the ML baselines, our approach is expected to be slightly slower because it adopts an iterative inferencing approach. But, this is compensated by solution accuracy, generalizability and robustness on high-resolution grids. A detailed analysis is provided in supplementary materials section E.

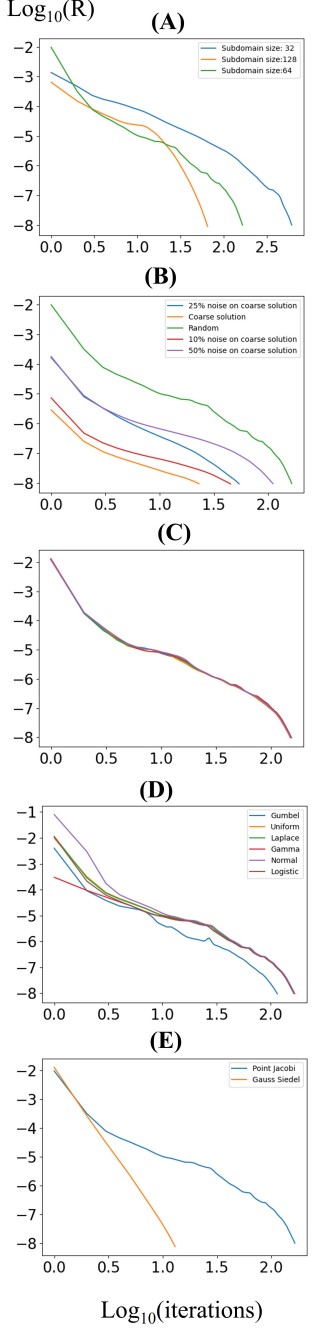

Figure 5: Convergence history for (A): varying subdomain sizes; (B): coupling with traditional PDE solvers; (C, D): Robustness and stability; (E): Point Jacobi vs Gauss Seidel

### 4.9 Importance of local learning and latent-space representation

To understand the effect of local learning, we set up the CoMLSim approach for a single subdomain (size equal to the computational domain). From this experiment, we conclude that the solution errors with a single subdomain is significantly higher than having multiple subdomains (See supplementary material section C.1.5.1). Additionally, to study the impact of latent space representations we experiment with the latent vector sizes as well as a case where CoMLSim solves in the solution space. We conclude that latent space representations have a larger impact on the computational cost than accuracy (See supplementary material section C.1.5.2).

### 4.10 Solution evolution during inference

In Figure. 6, we present the evolution of the PDE solution for the Poisson's equation experiment at different iterations of the iterative inferencing. The results are plotted for 3 unseen test cases with different source term distribution. In each case, the solution on the domain is initialized with random noise sampled from a uniform distribution between -1 and 1. The contour plot at iteration 0 shows an attenuated noisy solution because the initial solution is passed through the solution encoder and decoder. It may be observed that at iteration 1, the PDE solution starts evolving from the regions of high source term. As iterations progress, the solution begins to diffuse through the solution domain due to the repeated operation of the flux conservation network, until it converges. The diffusion process is dominant in the case of Poisson's equations and is effectively captured by the flux conservation autoencoder in CoMLSim. The error plot shows the solution accuracy at different iterations during the inference procedure. The log error is the largest initially and drops linearly as the iterations progress.

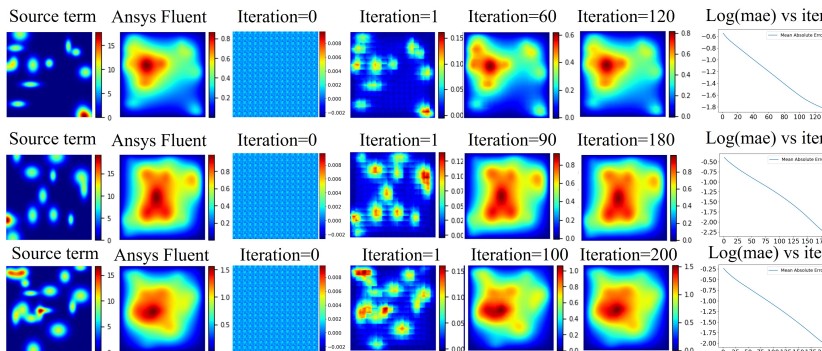

Figure 6: Evolution of PDE solution during inference

## 5 Conclusion

In this work, we introduce the CoMLSim approach, which is a self-supervised, low-dimensional and local machine learning approach for solving PDE on highly-resolved grids and generalizing across a wide range of PDE conditions. Our approach is inspired from strategies employed in traditional PDE solvers and adopts iterative inferencing. The proposed approach is demonstrated to predict accurate solutions for a range of PDEs, generalize reasonably across geometries, source terms and BCs. Moreover, it scales to bigger domains with larger mesh size.

**Broader impact, future work & limitations** Although the proposed ML-model can generalize to out-of-distribution geometries, source terms and BCs, but like other ML approaches, extrapolation to any and all PDE conditions that are significantly different from the original distribution still remains a challenge. However, this work takes a big step towards laying down the framework on how truly generalizable ML-based solvers can be developed. In future, we would like to address these challenges of generalizability and scalability by training autoencoders on random, application agnostic PDE solutions and enforcing PDE-based constraints in the iterative inferencing procedure. Future work will also investigate the potential for hybrid solvers and extensions to transient PDEs and inverse problems. Finally, we will also address extension of the current approach to unstructured meshes, which is a current limitation.

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
