# Supplementary Materials: A composable machine-learning approach for steady-state simulations on high-resolution grids

**Rishikesh Ranade**
Office of CTO
Ansys Inc.
Canonsburg, PA 15317
`rishikesh.ranade@ansys.com`

**Chris Hill**
Fluids Business Unit
Ansys Inc.
Lebanon, NH, 03766
`chris.hill@ansys.com`

**Lalit Ghule**
Office of CTO
Ansys Inc.
Canonsburg, PA, 15317
`lalit.ghule@ansys.com`

**Jay Pathak**
Office of CTO
Ansys Inc.
San Jose, CA, 95134
`jay.pathak@ansys.com`

In the supplementary materials, we provide additional details about our approach and to support and validate the claims established in the main body of the paper. We have divided the supplementary materials into 6 sections. Sections A and B provide details about the network architectures and training mechanics used in the CoMLSim approach as well as the ML baselines considered in this work. This is followed by additional experimental results in Sections C and D for PDEs considered in the main paper as well as additional canonical PDEs, namely Laplace and Darcy equations. Finally, we expand on the computational performance of CoMLSim in Section E and provide details of reproducibility in Section F.

## A    CoMLSim network architectures and training mechanics

In this section, we will provide details about the typical network architectures used in CoMLSim followed by the training mechanics. The training portion of the CoMLSim approach corresponds to training of several autoencoders to learn the representations of PDE solutions, conditions, such as geometry and PDE source terms as well as flux conservation. In this work, we mainly employ 2 autoencoder architectures, a CNN-based autoencoder to train the PDE solutions and conditions and a DNN-based autoencoder to train the flux conservation network.

### A.1    Solution/Condition Autoencoder

These autoencoders learn to represent solutions and conditions on subdomains into corresponding lower dimensional vectors. CNN-based encoders and decoders are employed here to achieve this compression because subdomains consist of structured data representations. Figure 1 shows the architecture of a typical autoencoder used in this work to learn PDE solutions and conditions. We use separate autoencoders to learn solution and representations of conditions into lower-dimensional latent vectors. In the encoder network, we use a series of convolution and max-pooling layers to extract global features from the solution. Irrespective of the size of the input, the pooling is carried out until a resolution of 4x4 in 2-D and 4x4x4 in 3-D. This is followed by flattening and a series of dense fully-connected layers to compute the latent vector. The decoder network mirrors the encoder network exactly, except that the pooling layer is replaced by an up-sampling layer. A ReLU activation function is applied after every convolution layer. The number of filters in the convolutional layers as

36th Conference on Neural Information Processing Systems (NeurIPS 2022).

well as number of dense layers and the bottleneck size depends on the complexity of the application, non-linearity and sparsity in the input distribution and the size of the subdomains.

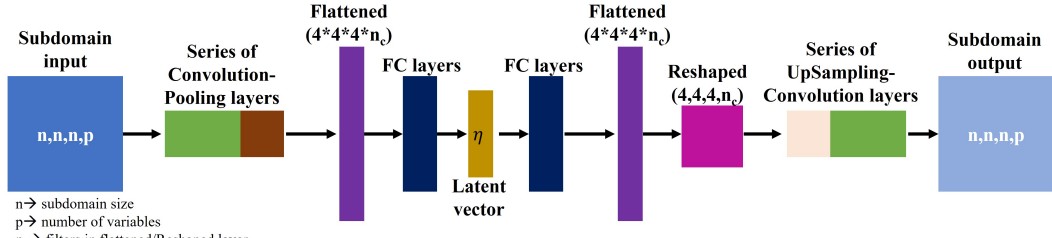

Figure 1: Schematic of Autoencoder network for solution and condition

### A.1.1  Input representation of PDE conditions

The experiments considered in this paper have different types of PDE conditions associated with the PDE. For example, Poisson's and Non-Linear Poisson's solutions are influenced by the source term, Reynolds Averaged Navier-Stokes external flow by geometry, Darcy's solutions by diffusivity and Laplace solutions by boundary conditions. Each PDE condition is encoded into a lower-dimensional vector using the autoencoder shown in Figure 1. Generally, diffusivity, source terms and boundary conditions have a spatial representation on the computational domain, which can be directly used to train the autoencoder. On the other hand, an efficient representation of geometry is a topic of on-going research in the ML community. Geometry can be represented using several ways such as point clouds, voxels, etc. In this work, we use the signed distance field (SDF) to represent geometry. Mathematically, the signed distance at any point within the geometry is defined as the normal distance between that point and closest boundary of a object. More specifically, for $x \in \mathrm{R}^n$ and object(s) $\Omega \subset \mathrm{R}^n$, the signed distance field $\phi(x)$ is defined by:

$$\phi(x) = \begin{cases} +d(x, \partial\Omega) & x \in \Omega \\ -d(x, \partial\Omega) & x \notin \Omega \end{cases}.$$

where, $\phi(x)$ is the signed distance field for $x \in \mathrm{R}^n$ and objects $\Omega \subset \mathrm{R}^n$ [1]. Maleki et al. [2] use the same representation of geometry to successfully demonstrate the encoding of geometries. However, this is a matter of choice and other valid representation can also be used in our approach.

All in all, there are two things to consider when encoding the PDE conditions, 1) the PDE condition is only encoded on subdomains that they influence and the encoding is hard-coded to a vector of zeros for all the other subdomains. For example, in an experiment of flow over a cylinder, the SDF is computed locally on each subdomain. The subdomain that cuts through the cylinder has a non-zero SDF and hence the encoding computed using the trained encoder is non-zero. Other subdomains that don't contain any parts of the cylinder can be encoded with a vector of zeros, 2) An autoencoder for PDE condition is required to be trained only if the set of conditions considered in a given problem have a spatial representation. If the PDE conditions are uniform, the magnitude can simply be considered as an encoding for a given subdomain. For example, if the source term is uniformly described on the computational domain for a given experiment, then the magnitude of the uniform source term can be used as an encoding on each subdomain.

### A.2  Flux conservation Autoencoder

These autoencoders learn to represent solution and condition encodings of a collection of neighboring subdomains. Since latent vectors don't have a spatial representation, DNN-based encoder and decoders are employed to compress them. Figure 2 shows a typical DNN-autoencoder used in this work to learn relationships between neighboring subdomains. The input to this autoencoder consist of PDE solution encoding ($\eta$) and condition encodings ($\eta_s, \eta_g, \eta_b$) on neighboring subdomains. In this work, the encoder network consists of typically 3 hidden layers with $n, n/2, n/4$ hidden neurons, respectively. The decoder network is similar to the encoder network but the order of the hidden layers is exactly opposite. The choice of $n$, depends on the size of the input vector and the complexity of the application. In the figure, we show an example of how the input to the flux conservation network

is setup for a 2-D case. Given the PDE solution and condition on a local stencil with 5 subdomains, the encoded representations are calculated using the pre-trained encoders. The encodings of solutions and conditions are concatenated together in a pre-determined order. The same approach works in 3-D, with the difference that the local stencil has 7 subdomains.

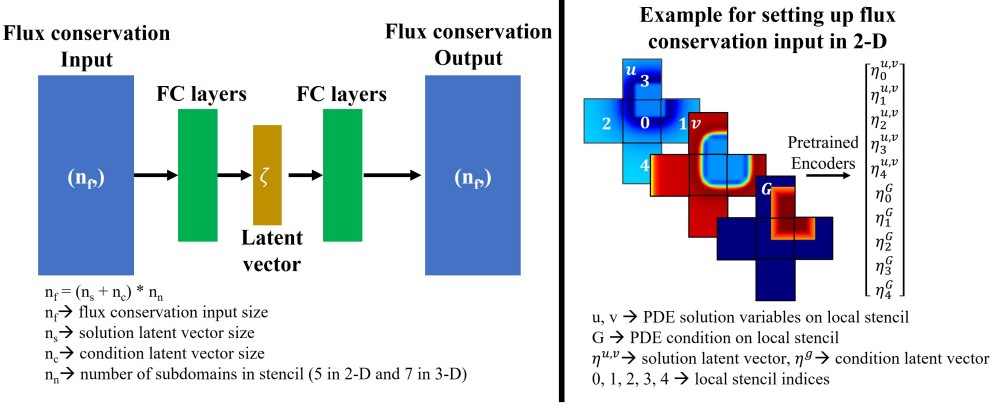

Figure 2: Schematic of Autoencoder network for flux conservation

## A.3 Training mechanics

Autoencoders have overfitting tendencies and hence they are required to be trained carefully. Here, we provide general guidelines that may be used to train these autoencoders efficiently. In this work, we train all the autoencoders until an MSE of $1e^{-6}$ or an MAE of $1e^{-3}$ is achieved on a validation set. More importantly, the compression ratio is selected such that the bottleneck layer has the smallest possible size and yet satisfies the accuracy up to these tolerances. Although, each training run is very fast but may require a decent amount of hyper-parameter tuning to obtain an optimized bottleneck size. Based on the experiments and results we have shown in the main paper, the optimum performance of our approach is observed in a range of bottleneck layer sizes. But, if the bottleneck size is too small or too large, the performance deteriorates. All the autoencoders are trained with the NVIDIA Tesla V-100 GPU using TensorFlow. The autoencoder training is a one-time cost and is reasonably fast.

## A.4 Similarities between CoMLSim and Traditional PDE solvers

In this section we expand on the main similarities in between our approach on a traditional Finite Volume or Finite Difference based PDE solvers. There are 3 main similarities, 1) Domain discretization, 2) Flux conservation and 3) Iterative solution algorithm. Here we provide more details about each one.

Consider a set of coupled PDEs with $n$ solution variables. For the sake of notation simplicity, we take $n = 2$, such that $u(x, y, z, t)$ and $v(x, y, z, t)$ are defined on a computational domain $\Omega$ with boundary conditions specified on the boundary of the computational domain, $\Omega_b$. It should be noted that extension to more solution variables is trivial. The coupled PDEs are defined as follows:

Figure 3: 2D stencil

$$L_1(u, v) - F_1 = 0; L_2(u, v) - F_2 = 0 \tag{1}$$

where, $L_1$, $L_2$ denote PDE operators and $F_1$, $F_2$ represent PDE source terms. The PDE operators can vary for different PDEs. For example, in a non-linear PDE such as the unsteady, incompressible Navier-Stokes equation the operator, $L = \frac{\partial}{\partial t} + \vec{a}.\vec{\nabla} - \vec{\nabla}.\vec{\nabla}$

1. **Domain discretization:** Traditional PDE solvers solve PDEs given in Eq. 1 by representing solutions variables, u, v, and their spatio-temporal derivatives on a discretized computational domain. The domain is discretized into a finite number of computational elements, using techniques such as Finite Difference Method (FDM), Finite Volume Method (FVM) and Finite Element Method (FEM).

Similar to traditional PDE solvers, the first step in the CoMLSim is to decompose the computational domain into smaller subdomains. A single subdomain in the CoMLSim is analogous to a computational element in the traditional solver because the CoMLSim predicts PDE solutions directly on local subdomains.

2. **Flux conservation:** Traditional PDE solvers use numerical approximation schemes are used to compute linear and non-linear components of the PDE. For example, in Eq. 1, if $L = \frac{\partial u}{\partial x} + \frac{\partial v}{\partial y}$, representing the 2-D incompressible continuity equation in fluid flows, the spatial derivatives can be approximated on a uniform stencil shown in Figure 3 using a second order Euler approximation shown below in Eq. 2.

$$L_{i,j} = \frac{u_{i+1,j} - u_{i-1,j}}{2\Delta x} + \frac{v_{i,j+1} - v_{i,j-1}}{2\Delta y} \qquad (2)$$

where $i, i+1, i-1, j, j+1, j-1$ are element indices and $\Delta x, \Delta y$ correspond to the size of stencil. These numerical approximations denote flux conservation between neighboring elements. Fluxes represent the flow of information between neighbors and hence, their accurate representation is crucial for information propagation within the domain.

Similarly, flux conservation in the CoMLSim happens across neighboring subdomains to ensure local consistency and information propagation. The representation of PDE discretization on subdomains is similar to Equation Eq. 2 but the indices i, j represent subdomain indices and the numerical schemes for discretization are now represented by a neural network, Theta, as shown in Eq. 3.

$$L_{i,j} = \Theta(\eta_{i,j}^u, \eta_{i+1,j}^u, \eta_{i-1,j}^u, \eta_{i,j+1}^u, \eta_{i,j-1}^u, \eta_{i,j}^v, \eta_{i+1,j}^v, \eta_{i-1,j}^v, \eta_{i,j+1}^v, \eta_{i,j-1}^v) \qquad (3)$$

where, $\eta_u, \eta_v$ are encodings of solution fields on subdomains and $i, i+1, i-1, j, j+1, j-1$ are subdomain indices.

3. **Iterative solution procedure:** In traditional solvers, the discretized PDEs represent a system of linear or non-linear equations, where the number of such equations equals the number of computational elements. To solve the PDE solutions, the discretized PDE residual is minimized by enforcing flux conservation iteratively using linear and non-linear equation solvers.

Similar to traditional solvers, the discretized PDEs represent a system of linear or non-linear equations, where the number of such equations equals the number of computational subdomains. To solve this system of equations we employ exactly the same techniques that a traditional would use. For example, in this work we have explored 2 linear iterative solution methods, such as Point Jacobi and Gauss Seidel.

### A.5  Self-supervised solution algorithm

Although, this approach requires solution samples to train the autoencoders, we claim in the paper that it is self-supervised in the sense that we don't use these samples to learn an explicit relationship between the input and output distribution. Our training consists of simply training autoencoders and the inference algorithm involves solving a constrained fixed-point iteration to converge to a PDE solution. In the constrained fixed-point iteration, the solution converges to a PDE solution starting from initial random noise. Our solution algorithm is never taught this trajectory of solution convergence but discovers that by itself. Hence, we claim that the solution algorithm at inference is self-supervised.

## B  Description of baseline network architectures

In the main paper, we compare the performance of CoMLSim with UNet [3], FNO [4], DeepONet [5] and FCNN [6]. Here we describe the network architectures used to train the respective models for all the experiments considered in the main paper.

**UNet [3]:** The encoder part of the network has 10 convolutional blocks, 2 at each down-sampled size. The input is down-sampled by 4x. The decoder part of the network predicts the output by up-sampling the bottleneck and using skip connections from the encoder network by concatenating the corresponding upsampled output with the corresponding down-sampled output. The decoder part

of the network also has 10 convolutional blocks, 2 after each up-sampled size and has 256 stacked channels. The total number of learnable parameters in UNet baseline is equal to 0.471 million in 2-D and 1.412 million in 3-D.

**Fourier Neural Operator (FNO) [4]:** The FNO model is same as the original implementation in Li et al. [4] but the number of modes are increased to 8 for 3-D experiments to achieve better training loss. The FNO model has 1.188 million parameters in 2-D and 3.689 million parameters in 3-D.

**DeepONet [5]:** The DeepONet architecture has two branches, a branch net and a trunk net. In all cases, the trunk net has 3 hidden layers with 512 neurons each. The branch is a convolutional neural network, which takes inputs the spatial source term. It is extremely difficult to train the DeepONet with the full resolution of the PDE conditions because of the massive data storage requirements. Hence, for all experiments the PDE conditions are uniformly subsampled to a lower grid resolution given as an input to the branch net. The branch net has 4 DownSample blocks and 10 convolutional blocks, 2 at each down-sampled size. Additionally, the DeepONet is extremely sensitive to the sampling strategy adopted in the training data. The total number of learnable parameters is equal to 1.353 million. The subsampling of PDE conditions and the random sampling used in this work may have affected the testing accuracy of DeepONet.

**Fully Convolutional Neural Network (FCNN) [6]:** The FCNN model is similar to the original implementation in Zhu and Zabaras [6] but the number of convolution filters and downsampling layers are tuned to accommodate the high-resolutions and non-linearity in different use cases. The FCNN model has 0.189 million parameters in 2-D and 0.578 million parameters in 3-D.

## C   Experiments results and details from main paper

We demonstrated the CoMLSim for 4 experiments in the main paper. Here we provide more details about the CoMLSim setup as well as additional results and discussions for each experiment. The different experiments presented in this work are a good mix of pure research and engineering problems with varying levels of non-linear complexity, input distributions, PDEs, solution variable coupling, spatial dimension etc. It must be noted that CoMLSim as well as the baselines are trained with reasonably training samples. The baseline performance can have different outcomes with increasing the size of the training data.

### C.1   2-D Linear Poisson's equation

The Poisson's equation is shown below in Eq. 4.

$$\nabla^2 u = f \tag{4}$$

where, $u$ is the solution variable and $f$ is the source term. In this experiment, the source term is sampled from a Gaussian mixture model, where the number of Gaussians is randomly chosen between 1 and 30 and each Gaussian has a randomly specified mean and standard deviation. The Gaussian mixture model is described below in Eq. 4. The computational domain is 2D and is discretized with a highly-resolved grid of resolution 1024x1024. The high-resolution grid is required in this case to capture the local effects of the source term distribution.

$$\sum_{j=0}^{1024}\sum_{j=0}^{1024} f_{i,j} = \sum_{j=0}^{1024}\sum_{j=0}^{1024}\sum_{k=0}^{30} A_k \exp\left(-\frac{x-\mu_{x,k}}{\sigma_{x,k}} - \frac{y-\mu_{y,k}}{\sigma_{y,k}}\right) \tag{5}$$

where, $x, y$ correspond to the grid coordinates. $A_k$ randomly assumes either 0 or 1 to vary the number of active Gaussians in the model. $\mu_x, \mu_y$ and $\sigma_x, \sigma_y$ are the mean and standard deviations of Gaussians in $x$ and $y$ directions, respectively. The means vary randomly between 0 and 1, while the standard deviations are varied between 0.001 and 0.01. The smaller magnitude of standard deviation results in hot spots that can only be captured on highly-resolved grids.

### C.1.1   Training

256 solutions are generated for random Gaussian mixtures using Ansys Fluent and used to train the different components of CoMLSim. The computational domain of 1024x1024 resolution is divided

into 256 subdomains each of resolution 64x64. The solution and source terms are compressed into latent vectors of size 11, respectively. The flux conservation autoencoder has a bottleneck layer of size 35.

### C.1.2 Testing

100 more solutions are generated for random Gaussian mixtures using Ansys Fluent. Due to the high-dimensionality of the source term description, the testing set has no overlap with the training set. The convergence tolerance of the CoMLSim solution algorithm is set to $1e^{-8}$ and the solution method is Point Jacobi. Each solution converges in about 2 seconds and requires about 150 iterations on an average.

### C.1.3 Comparisons with Ansys Fluent for in-distribution testing

Source term  Ansys Fluent  CoMLSim

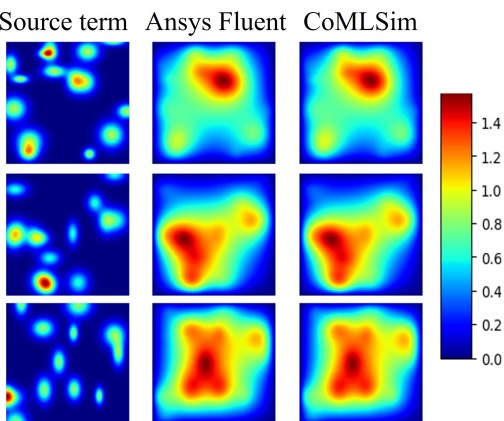

Figure 4: CoMLSim vs Ansys Fluent for linear Poisson's equation with unseen in-distribution source terms

The CoMLSim predictions for a selected unseen testing samples are compared with Ansys Fluent in Fig. 4. It may be observed that the contour comparisons agree well with Fluent solutions. Overall, the mean absolute error over 100 testing samples is 0.011.

### C.1.4 Comparisons with Ansys Fluent for out-distribution testing

In this section, we demonstrate the generalization capability of CoMLSim as compared to Ansys Fluent and all the baselines. We have not presented results from DeepONet because it is does not perform as well as the other baselines. We consider 3 experiments outlined below.

1. **Higher number of Gaussians:** It may be observed from Eq. 5 that the maximum number of Gaussians allowed is set to 30 and this number is sampled from a uniform distribution. In this experiment, we fix the number of Gaussians to 30, 40, 50 and 60 and generate 10 solutions for each case using Ansys Fluent. As the number of Gaussians increase the source term distribution moves further away from the distribution in Eq. 5 used in training.

   It may be observed from the contour plots in Figure 5, that the CoMLSim approach consistently beats all the ML baselines. As the number of Gaussians increases, the total source term applied on the computational domain increases proportionally. As a result, the magnitude of the solution is also higher and this can be observed in the color map scale in Figure 5. The CoMLSim captures the both the spatial solution pattern as well as the magnitude correctly in comparison to all ML-baselines. Amongst the ML baselines, FNO performs the best.

2. **Completely different source distribution:** In this experiment, we sample the source term from a completely different distribution. The source term distribution is discontinuous, where the computational domain is divided into either $8^2$ or $16^2$ tiles and a uniform source term is specified on each tile such that the total source is conserved. The resulting discontinuity in source terms between neighboring tiles makes it challenging to calculate the solution, even

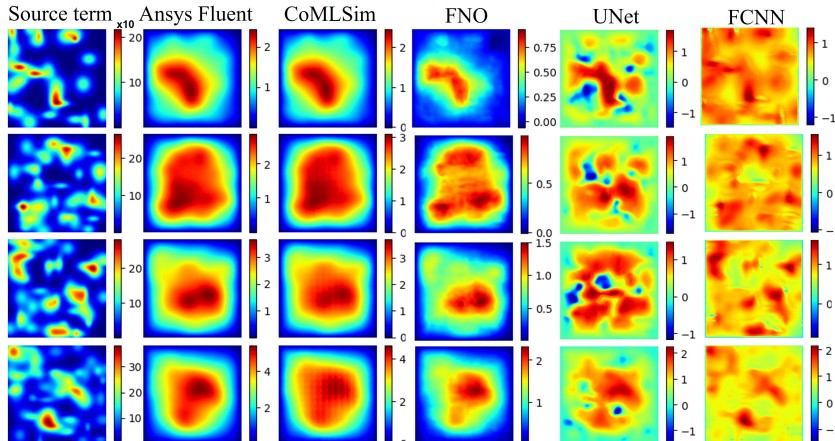

Figure 5: Comparison for linear Poisson's equation on out-of-distribution source terms. Number of Gaussians vary from 30 to 60 from top to bottom.

with traditional PDE solvers. We generate 10 solutions each with $8^2$ and $16^2$ tiles using Ansys Fluent. This source term distribution is significantly different from the Gaussian distribution that was initially considered for training.

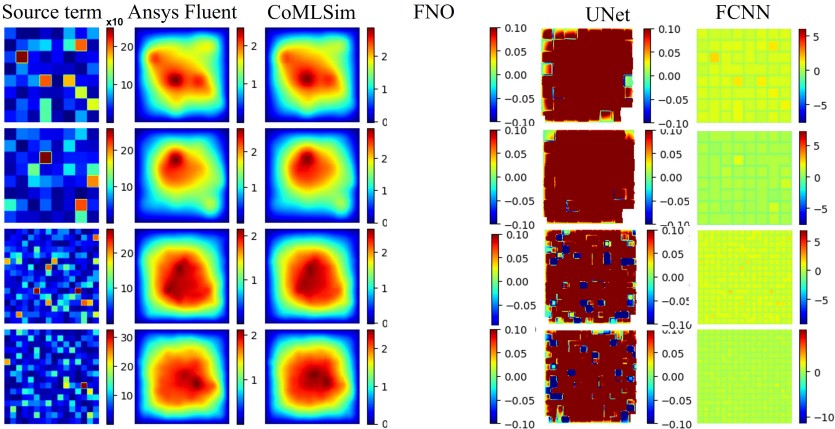

Figure 6: Comparison for linear Poisson's equation on out-of-distribution discontinuous source terms

It may be observed from the contour plots in Figure 6, that the ML baselines don't predict a reasonable solution. On the other hand, the CoMLSim predicts solutions with higher accuracy. This can be attributed to the local and latent space learning strategies adopted which enables it to accurately learn and predict the local physics of the PDE. It may also be observed that FNO predicts NaN values because the Fourier transform of the discontinuous source term in the first layer of FNO results in NaN.

3. **Bigger domain with mesh size of 2048x2048:**

   In the main paper, we showed how CoMLSim is designed to scale to bigger physical domains with larger mesh sizes. Here, we present comparisons at a mesh size of $2048^2$ between CoMLSim, Ansys Fluent and the ML baselines for the first 4 samples in the testing set. It may be observed from Figure 8 that the CoMLSim outperforms all baselines and matches well with Ansys Fluent.

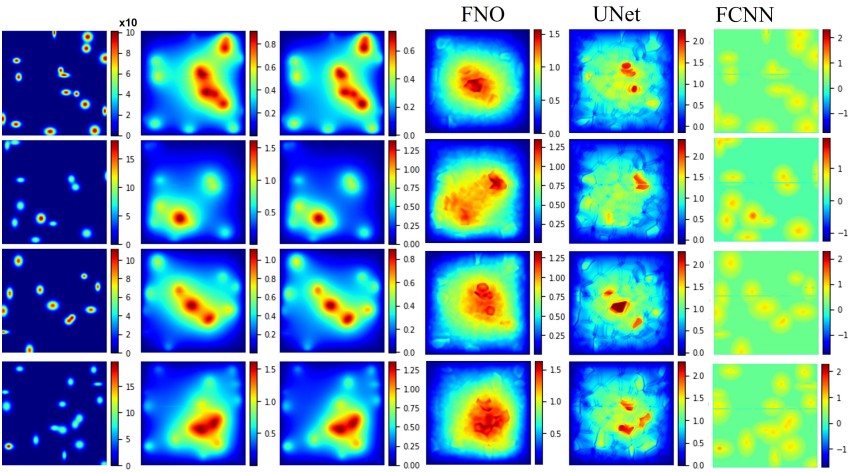

Figure 7: Comparison for linear Poisson's equation for a bigger domain with mesh size of 2048x2048

### C.1.5 Additional Ablation Studies

In this section we carry out additional ablation studies to further understand the different aspects of the CoMLSim approach. The CoMLSim has 3 main features, 1) Local learning, 2) Latent space representations and 3) iterative inferencing. The ML components in our approach, such as the solution, condition and flux conservation autoencoders are designed to support these features. In this section, we present experiments to understand the impact of each of these features on the performance of our approach.

**1) Importance of local learning:**

In this experiment we setup the CoMLSim with a single subdomain. The solution and source term autoencoders, in this case, are trained on the entire computational domain (1024x1024). Since there are no neighbors, the flux conservation autoencoder is trained on a 1-subdomain stencil. The CoMLSim approach is tested on 250 samples. The mean absolute error obtained is equal to 0.072. The cases reported in the paper in Section 4.3 with 32, 64 and 128 subdomains have mean absolute errors of 0.015, 0.011 and 0.029, respectively. In comparison to Table 1 from the paper, it may be observed that the errors from the single subdomain CoMLSim approach are similar to the ML-baselines. On the other hand, the computational time required by the single subdomain instance is about 3x more expensive than multi-subdomain instance of CoMLSim. This is due to large latent vector sizes obtained from encoding solutions on highly resolved computational domains (1024x1024) as opposed to using subdomains (64x64). Moreover, single subdomain CoMLSim instance cannot scale to bigger computational domains with larger meshes.

**2) Importance of latent space representation:**

In this experiment, we set the encoder and decoder of solution and condition autoencoders to identity. By doing this we show that as we reduce to a standard domain decomposition, the accuracy of the approach is maintained but the computational time significantly increases. As stated earlier, we use a subdomain resolution of 64x64 to discretize the domain of 1024x1024 resolution into 256 smaller subdomains. With identity encoders and decoders, the latent size of solutions and source terms on each subdomain is equal to 4096, which is the number of computational elements in a subdomain. As a result, the flux conservation autoencoder has an input of (4096 + 4096) * 5 = 40960, where 5 is the number of surrounding neighbors. Training a fully connected neural network with 40960 input features requires networks with billions of parameters and computationally prohibitive GPU memory. Due to these reasons, we reduce the subdomain size to 16x16 to generate a comparison for accuracy. The flux conservation compression ratio (bottleneck size / input size) is consistent with the experiment in the paper. The accuracy observed on 250 testing samples is around 0.017 and is consistent with the results reported Table 1. However, the computational time required in this case is about 2-3x larger than the results reported in Section E.

**3) Impact of latent vector size:**

In this experiment we evaluate the impact of solution and source term latent sizes with accuracy and computational time of CoMLSim approach averaging over 250 testing samples. It may be observed from the table 1 that the accuracy decreases as we increase the latent sizes. This is due to overfitting of the autoencoder, which makes the latent vector less pronounced. The computational iterations also increase. This observation is consistent with the experiment in Section 4.4 of the main paper where we vary the size of the latent size in flux conservation autoencoder.

It is important to note that the result of this experiment cannot be compared to standard domain decomposition. The solution and condition autoencoders in the case of standard domain decomposition were identity and resemble error-free encoding. In this case, autoencoders are overfit for larger latent sizes and substantially diminish the prominence of the latent vector, thereby resulting in bad performance.

Table 1: Solution/Condition Encoding size vs accuracy and number of convergence iterations

| Encoding size | Mean Absolute Error | Avg. Iterations |
|---|---|---|
| 7 | 0.047 | 75 |
| 11 | 0.017 | 150 |
| 32 | 0.029 | 250 |
| 64 | 0.1 | 600 |
| 128 | 0.21 | 700 |
| 256 | 0.29 | 800 |

## C.2    2-D Non-linear coupled Poisson's equation

The coupled non-linear Poisson's equation is shown below in Eq. 6.

$$\left. \begin{aligned} \nabla^2 u &= f - u^2 \\ \nabla^2 v &= \frac{1}{u^2 + \epsilon} - v^2 \end{aligned} \right\} \tag{6}$$

where, $u, v$ are the solution variables and $f$ is the source term. The source term is similar to the experiment in Section C.1 except that the $x, y$ standard deviations are varied between 0.001 and 0.05. As opposed to linear Poisson's, in this problem the complexities result from the coupling of solution variables as well as the source term distribution. A closer look at the PDE in Eq. 6 shows us that the variable $v$ is implicitly coupled with the source term and this relationship is challenging to discover for ML methods.

### C.2.1    Training

256 solutions are generated for random Gaussian mixtures using Ansys Fluent and used to train the different components of CoMLSim. The computational domain is divided into 256 subdomains each of resolution 64x64. The solution and source terms are compressed into latent vectors of size 11, respectively. The flux conservation autoencoder has a bottleneck layer of size 35.

### C.2.2    Testing

100 more solutions are generated for random Gaussian mixtures using Ansys Fluent. The convergence tolerance of the CoMLSim solution algorithm is set to $1e^{-8}$ and the solution method is Point Jacobi. Each solution converges in about 2 seconds and requires about 150 iterations on an average.

### C.2.3    Comparisons with Ansys Fluent for in-distribution testing

The CoMLSim predictions for a selected unseen testing samples are compared with Ansys Fluent in Fig. 8. It may be observed that the contour comparisons agree well with Fluent solutions. Overall, the mean absolute error over 100 testing samples is 0.0053. The mean absolute error is smaller than the linear Poisson experiment because of the smoothness introduced in the power map by increasing the Gaussian mixture standard deviation. Comparisons with other ML baselines are provided in the main paper.

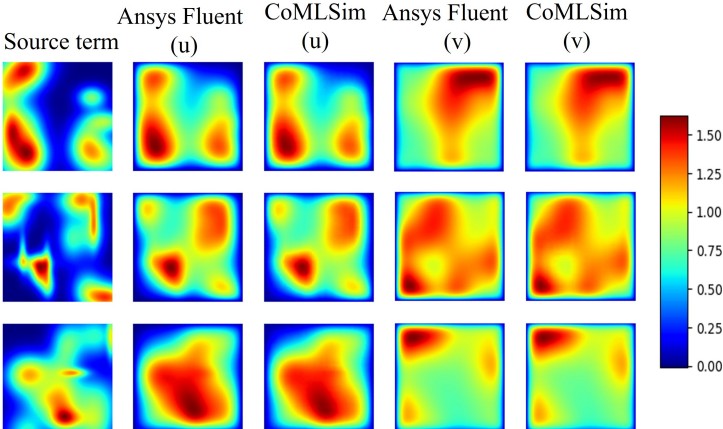

Figure 8: CoMLSim vs Ansys Fluent for non-linear Poisson's equation for in-distribution unseen source terms

### C.2.4 Comparisons with Ansys Fluent for out-distribution testing

In this section, we demonstrate the generalization capability of CoMLSim as compared to Ansys Fluent and all the baselines. Similar to the generalizability test with linear Poisson's, we fix the number of Gaussians to 30, 40, 50 and 60 and generate 10 solutions for each case using Ansys Fluent. As the number of Gaussians increase the source term distribution moves further away from the distribution in Eq. 5 used in training.

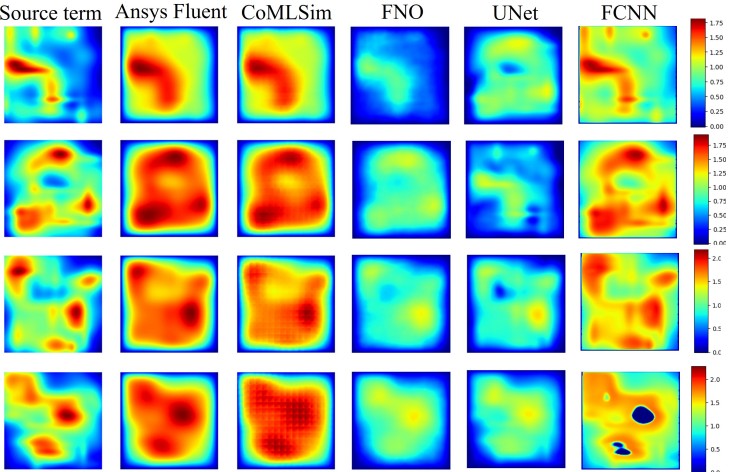

Figure 9: Comparison for variable $u$ of non-linear Poisson's equation for out-of-distribution source terms. The number of Gaussians vary from 30 to 60 from top to bottom.

It may be observed from Figures 9 and 10 that the CoMLSim outperforms all the ML baselines and matches well with Ansys Fluent for both solution variables. The out-of-distribution source term has a much stronger impact on the coupled solution variable as compared to linear Poisson's experiment. It may be observed from the figures that the ML-baselines predict reasonably for solution variable $u$, but the errors are much higher for the coupled variable, $v$.

### C.3  3-D Reynolds Averaged Navier-Stokes external flow

In this experiment, we apply the CoMLSim approach to model high Reynolds number turbulent flow around arbitrary geometry shapes built for primitive objects such as Cylinder, Cuboid, NURBS etc and their combinations. The governing PDEs for this case are shown in Eq.8

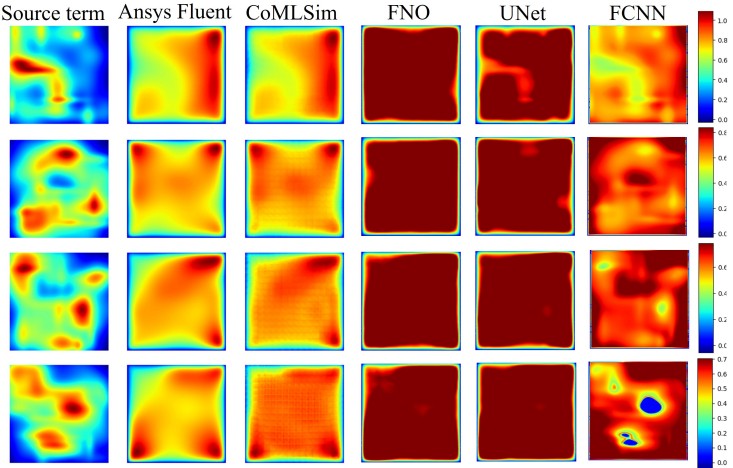

Figure 10: Comparison for variable $v$ of non-linear Poisson's equation for out-of-distribution source terms. The number of Gaussians vary from 30 to 60 from top to bottom.

$$
\left.
\begin{aligned}
&\textbf{Continuity:} &&\nabla.\mathbf{v} = 0 \\[4pt]
&\textbf{Momentum:} &&(\mathbf{v}.\nabla)\mathbf{v} + \nabla\mathbf{p} - \nabla.(\nu_t\nabla\mathbf{v}) = 0 \\[4pt]
&\textbf{TKE:} &&(\mathbf{v}.\nabla)\mathbf{k} - \nabla.\left(\frac{\nu_t}{\sigma_k}\nabla\mathbf{k}\right) - 2\nu_t E.E + \mathbf{e} = 0 \\[4pt]
&\textbf{TDR:} &&(\mathbf{v}.\nabla)\mathbf{e} - \nabla.\left(\frac{\nu_t}{\sigma_e}\nabla\mathbf{e}\right) - C_{1e}\frac{\mathbf{e}}{\mathbf{k}}2\nu_t E.E + C_{2e}\frac{\mathbf{e}^2}{\mathbf{k}} = 0
\end{aligned}
\right\} \quad (7)
$$

where, $k$ is Turbulent Kinetic Energy (TKE), $e$ is Turbulent Dissipation Rate (TDR), $v = (u_x, u_y, u_z)$ is the velocity vector, $p$ is pressure, $\nu_t$ is the kinematic turbulent viscosity, $C_{1e}, C_{2e}, \sigma_k, \sigma_e$ are empirical constants. More details related to this model and approach can be found in Alfonsi [7]. Modeling of Reynolds averaged turbulence presents an additional challenge because it involves 2 additional solution variables which are strongly coupled and influence the velocity and pressure of the flow.

The use case consists of a 3-D channel flow over arbitrarily shaped objects with a computational domain that has a grid resolution of 304x64x64. The object is placed on the bottom surface of the channel to simulate conditions of automobiles moving on a road. The domain has a velocity inlet specified at 40m/s and a zero pressure outlet boundary condition, while the rest of the surfaces are walls with no-slip conditions. The high resolution grid is required because the large flow Reynolds number, of the order of 10000, results in large velocity gradients which can affect the entire flow field. In this experiment, we vary only the geometric features of the object but in future we would like to develop an external flow solver that encompasses both geometry and Reynolds number variations.

### C.3.1 Training

150 solutions are generated using Ansys Fluent for geometries corresponding to 5 primitive objects such as cylinder, cuboid, trapezoid, NURB and wedge and their random rotations along all axes as well as their combinations. The computational domain of 304x64x64 is divided into smaller subdomains each with a resolution 8x8x8. The solutions and geometry are encoded to a latent vectors of size 64 and 32, respectively. The geometry is represented with a signed distance field in each subdomain. The flux conservation network has a bottleneck layer of size 64.

### C.3.2 Testing

50 additional solutions are generated using Ansys Fluent for other geometries sampled from the same distribution. These geometries are not a part of of the training set. The convergence tolerance of the

CoMLSim solution algorithm is set to $1e^{-8}$ and the solution method is Gauss Seidel. Each solution converges in about 35 seconds and requires about 75 iterations on an average.

### C.3.3 Comparison with Ansys Fluent for geometries outside the training set

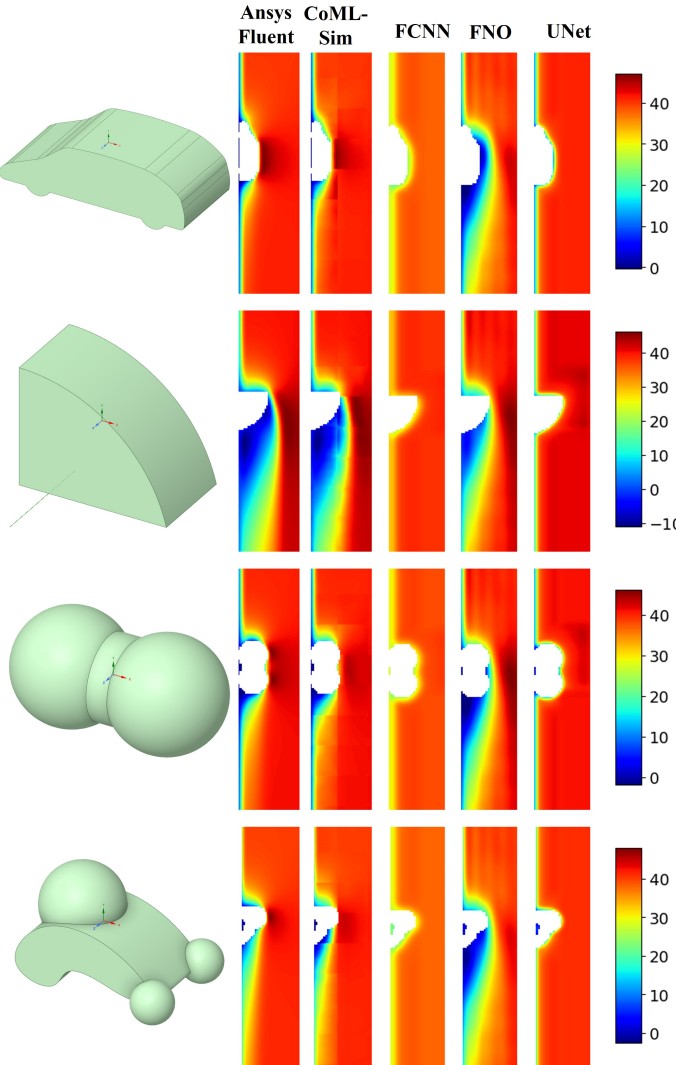

Figure 11: X-velocity contour comparisons (Figures rotated to fit on page and zoomed in on the object)

It may be observed from Figures 11 and 12 that the CoMLSim predictions match well with Ansys Fluent for both x-velocity and pressure solutions. The geometries considered in this experiment were not a part of the training set and can be considered out-of-distribution for CoMLSim and other ML baselines. The different features of the flow field are captured reasonably well. As the flow first hits the object it severely decelerates causing smaller velocities and higher pressures. This is known as the stagnation point. Beyond this point, the flow travels along the surface of the object and eventually separates from the object leaving a recirculating wake on the downstream of the object. The point of separation is dependent on the geometry of the object and the flow speed, and determines the pattern of the recirculating wake. As the flow separates, it begins to accelerate resulting in a drop in pressure. It may be observed from the figure that all of these flow features are captured accurately by CoMLSim for the objects considered in this experiment. On the other hand, the other ML baselines do not capture the flow physics accurately. The error comparisons of CoMLSim and other ML baselines in comparison to Ansys Fluent are reported in the table 2. It may be observed that CoMLSim outperforms the baselines.

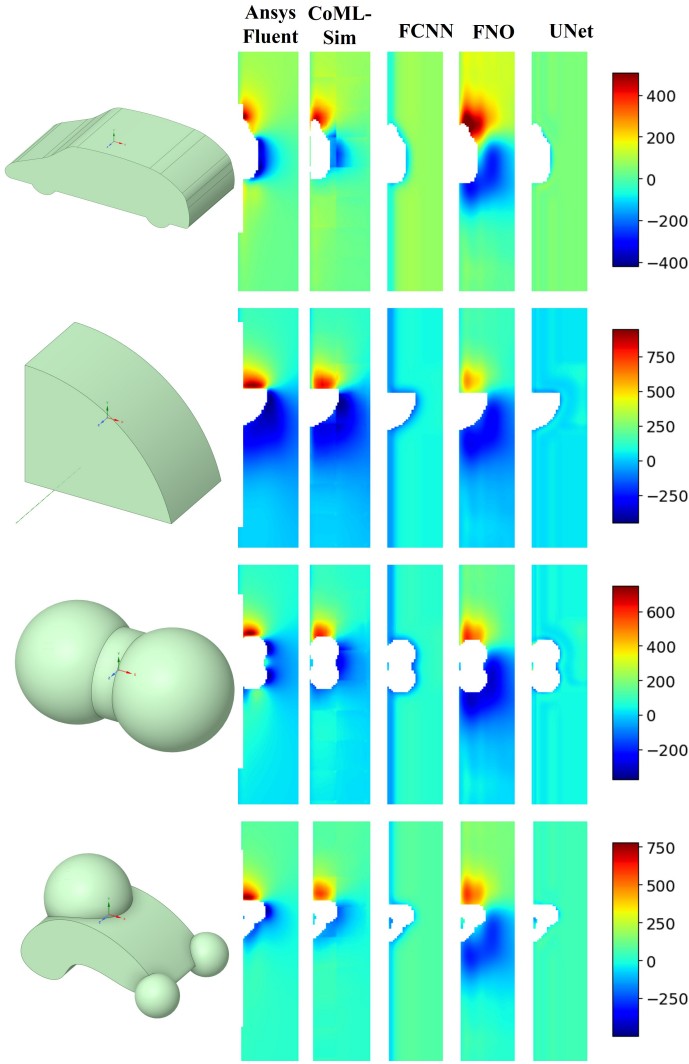

Figure 12: Pressure contour comparisons (Figures rotated to fit on page and zoomed in on the object)

Table 2: Mean Abs. Errors normalized by inlet velocity (40 m/s) in X-Velocity and Pressure

|  | X-Velocity | Pressure |
|---|---|---|
| CoMLSim | 0.02025 | 0.088 |
| UNet | 0.0435 | 1.178 |
| FNO | 0.0315 | 0.5502 |
| FCNN | 0.1395 | 1.6685 |

Amongst the baselines, FNO performs the best and this is evident from the contours in Figures 11 and 12 as well as the errors reported in table 2. On the other hand, FCNN and UNet are severely affected by the sparsity of the PDE solutions observed in all the flow variables and hence don't perform well. We would like to clarify that all the ML baselines were trained to the best of our ability. We provide loss curves in Figure 13 for the benefit of the reader to show that each baseline is trained until the training and validation losses plateau. Additionally, it may also be observed the loss drops by 3 orders of magnitude for each baseline. Moreover, as described in Section B, the networks have a sufficient number of parameters to train efficiently. Finally, traditional ML methods, such as the ones considered in this paper, are data-hungry and improve results as more data becomes available. However, in this experiment, we trained all the models with only 150 solution samples on highly-resolved grids and this reflects in the poor solution accuracy of these methods. Increasing

the depth of the networks and number of parameters to 5.38 million for UNet and 2.33 million for FCNN did not improve the results.

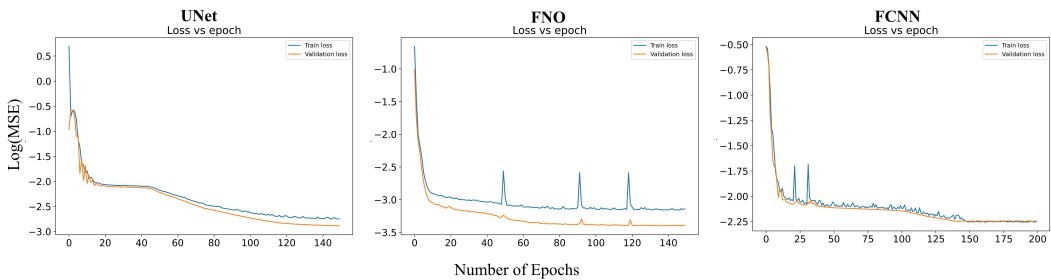

Figure 13: Training loss curves for all the ML baselines

### C.3.4    Extension to larger physical domains and complex geometries

In this section, we demonstrate the scalability of the CoMLSim approach to bigger physical domains with substantially higher mesh sizes. The results presented previously are carried out on physical domains with grid resolutions of 304x64x64. In this experiment, we increase the grid size to 562x128x128 by increasing the physical size of the domain as well as the geometry. Additionally, we use an unseen test geometry of a car model to perform this evaluation.

Figure 14 shows the streamlines and vector plots of fluid flow near the surface of the car. When compared with Ansys Fluent we observe reasonable mean absolute errors of 0.039 for velocity and 0.106 for pressure. Moreover, it is important to note that the performance and scalability of our approach to larger meshes is not affected by the 3-D spatial representation in this use case. Most ML baselines considered in this work run out of computational memory when evaluating on this mesh.

**Norm. MAE on velocity = 0.039**
**Norm. MAE on pressure = 0.106**

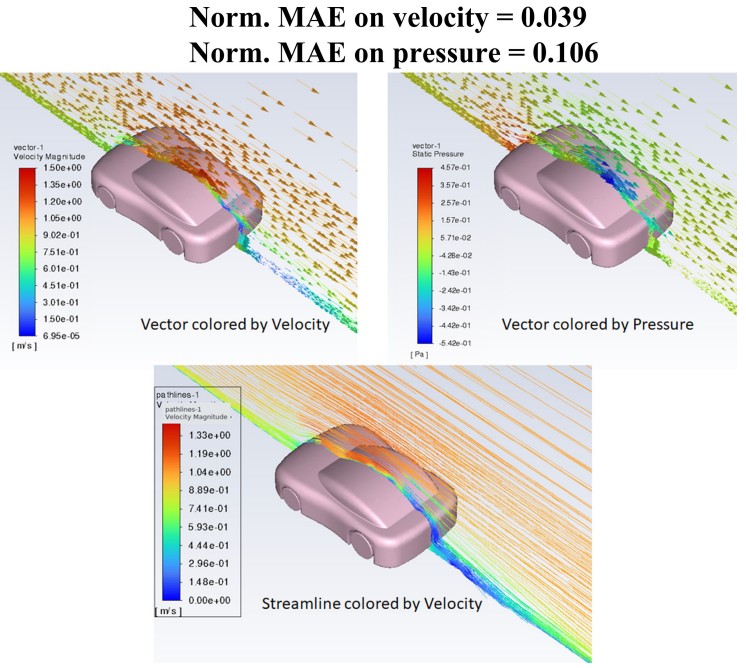

Figure 14: CoMLSim evaluation on bigger domains and unseen car model

## C.4 Industrial usecase: 3-D electronic chip cooling with natural convection

In this section, we demonstrate the CoMLSim approach for solving the complicated industrial use case of electronic chip cooling with a wide range of applications. The main purpose of presenting this experiment is to show the ability of our approach can be used for solving real industrial application with similar accuracy's as traditional PDE solvers.

In this case the domain consists of a chip, which is sandwiched in between an insulated mold. The chip-mold assembly is held by a PCB and the entire geometry is placed inside a fluid domain. The geometry and case setup of the electronic chip cooling case can be observed in Fig. 15. The chip is subjected to electric heating and the uncertainty in this process results in random spatial distribution of heat sources on the on the surface of the chip. The power map distribution in this case sampled from the same Gaussian mixture model described in Eq. 5, but the total number of Gaussians is limited to 15.

The physics in this problem is natural convection cooling where the power source is responsible for generating heat on the chip, resulting in an increase in chip temperature. The rising temperatures get diffused in to the fluid domain and increase the temperature of air. The air temperature induces velocity which in turn tries to cool the chip. At equilibrium, there is a balance between the chip temperature and velocity generated and both of these quantities reach a steady state. The objective of this problem is to solve for this steady state condition for an arbitrary power source sampled from a Gaussian mixture model distribution, which is extremely high dimensional. The governing PDEs that represent this problem are shown in Eqs. 8.

$$
\left.
\begin{aligned}
&\textbf{Continuity:} && \nabla.\mathbf{v} = 0 \\
&\textbf{Momentum:} && (\mathbf{v}.\nabla)\mathbf{v} + \nabla\mathbf{p} - \frac{1}{Re}\nabla^2\mathbf{v} + \frac{1}{\beta}\vec{g}\mathbf{T} = 0 \\
&\textbf{Heat (Solid):} && \nabla.(\alpha\nabla\mathbf{T}) - P = 0 \\
&\textbf{Energy (Fluid):} && (\mathbf{v}.\nabla)\mathbf{T} - \nabla.(\alpha\nabla\mathbf{T}) = 0
\end{aligned}
\right\} \tag{8}
$$

where, $v = u_x, u_y, u_z$ is the velocity field in $x, y, z$, $p$ is pressure, $T$ is temperature, $Re$ and $\alpha$ are flow and thermal properties, $P$ is the heat source term, $\frac{1}{\beta}\vec{g}T$ is the buoyancy term. $P$ is the spatially varying power source applied on the chip center. The main challenges are in capturing the two-way coupling of velocity and temperature and generalizing over arbitrary spatial distribution of power.

The coupled PDEs with 5 solution variables, $v = u_x, u_y, u_z, P, T$ are solved on a fluid and solid domain with loose coupling at the boundaries. The fluid domain is discretized with $256^3$ elements in the domain and the solid domain (chip) is modeled as a 2-D domain with $64^2$ elements as it is very thin in the third spatial dimension.

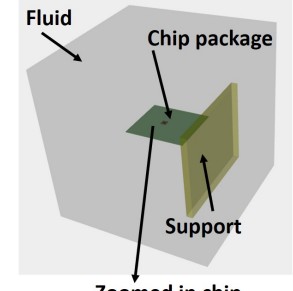

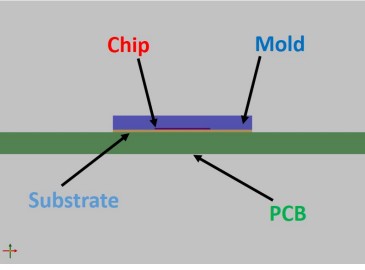

### C.4.1 Training

The data to train the autoencoders in the CoMLSim approach is generated using Ansys Fluent and corresponds to 300 PDE solutions. The computational domain is divided into 4096 subdomains, each with $16^3$ computational elements. The solution autoencoders for the 5 solution variables are trained to establish lower dimensional latent vectors with size 29 on the subdomain level.

### C.4.2 Testing

100 more solutions are generated for random Gaussian mixtures using Ansys Fluent. The convergence tolerance of the CoMLSim solution algorithm is set to $1e^{-8}$ and the solution method is Gauss Seidel. Each solution converges in about 42 seconds on an average.

Figure 15: Electronic chip cooling geometry

### C.4.3 Comparison with Ansys Fluent

**Contour plots across different metrics:**

The CoMLSim predictions for 2 randomly selected unseen testing samples are compared with Ansys Fluent in Figs. 16 and 17. It may be observed that the contour comparisons agree well with Fluent solutions and the mean absolute errors are of the order of $1e^{-2}$ for temperature and $1e^{-3}$ for velocity. Similar results are observed for other power maps from the unseen testing set.

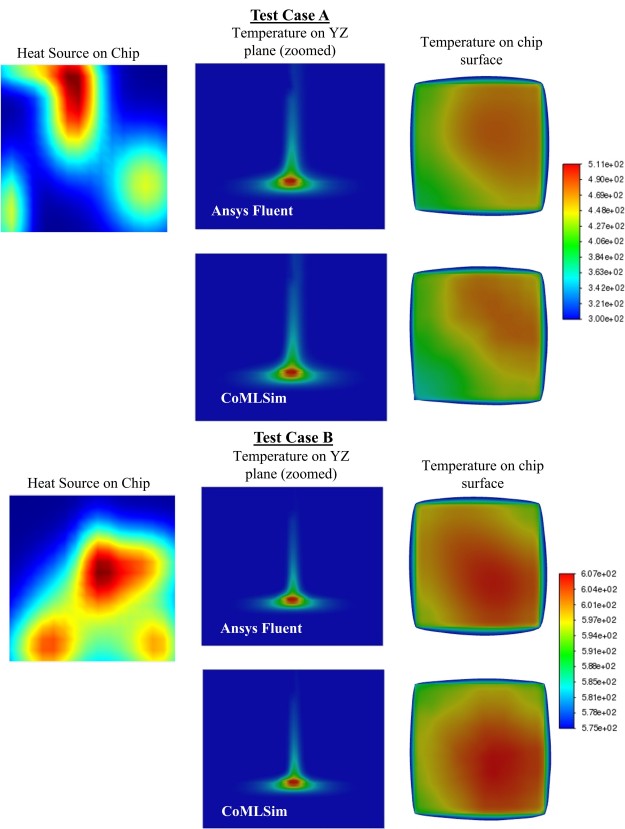

Figure 16: Temperature contour comparisons for unseen powermaps

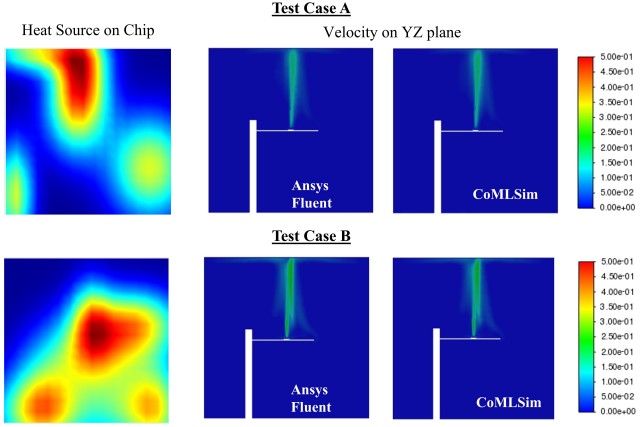

Figure 17: Velocity contour comparisons for unseen powermaps

**Line plots across different metrics:**

Figures 18 and 19 shows the line comparisons of temperature and velocity plotted along the chip across all three spatial directions. These metrics provide a more quantitative representation of the accuracy of our approach. It maybe observed that the predictions of CoMLSim agree well with Ansys Fluent for the 2 test cases.

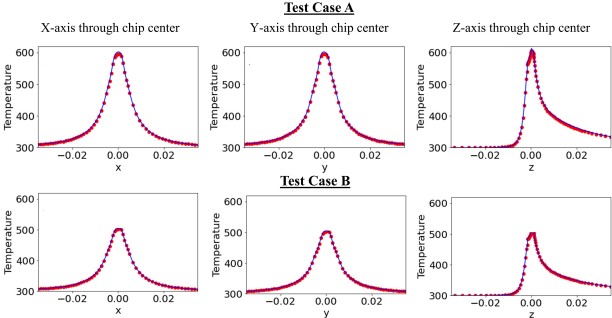

Figure 18: Line comparisons of temperature for unseen powermaps (blue:CoMLSim, red:Ansys Fluent

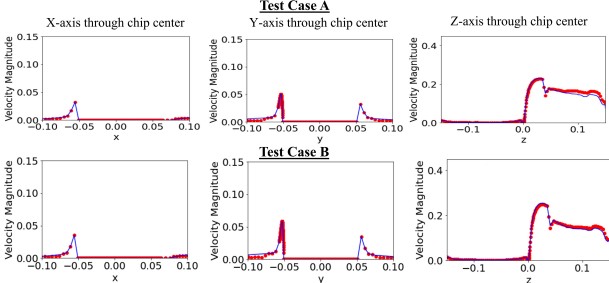

Figure 19: Line comparisons of velocity for unseen powermaps (blue:CoMLSim, red:Ansys Fluent

**Comparisons with ML baselines for other quantitative metrics:**

Table 3: Testing errors for different metrics

|  | $L_\infty(T)$ | $\epsilon(T_{max})$ | $\epsilon(heat\ flux)$ |
|---|---|---|---|
| CoMLSim | 15.2 | 5.09 | 1.26 |
| UNet | 95.21 | 76.2 | 27.56 |
| FNO | 60.83 | 5.97 | 12.12 |
| FCNN | 192.7 | 191.38 | 311.23 |

Finally, in Table 3 we present 3 different testing metrics for the CoMLSim predictions, 1) Error in maximum temperature in computational domain (hot spots on chip), 2) $L_\infty$ error in temperature in the computational domain, and 3) Error in heat flux (temperature gradient) on the chip surface. These metrics are more suited for this application and provide a much better measure for evaluating accuracy. All errors are computed relative to the Ansys Fluent solution and are averaged over 100 the unseen testing samples. It may be observed from Table 3 that CoMLSim agrees well with Ansys Fluent on all the comparison metrics and outperforms all the ML baselines.

# D    Experiments for additional canonical PDEs

## D.1    2-D Laplace Equations:

The Laplace equation represents a canonical problem for benchmarking linear solvers. It is shown below in Eq. 9.

$$\nabla^2 \phi = 0 \tag{9}$$

where, $u$ is the solution variable subjected to a Dirichlet boundary condition, $\phi(\vec{x_b}) = f_b$ or a Neumann boundary condition, $\frac{\partial \phi}{\partial \vec{x_b}} = f_b$. The boundary conditions (BCs) are sampled randomly and the magnitude of the BC is uniformly sampled between $-1.0$ and $1.0$.

### D.1.1 Training

300 solutions are generated for arbitrary BCs using Ansys Fluent on a computational domain with a resolution of 256x256. These solutions are used to train the different components of CoMLSim. The computational domain is divided into $64$ subdomains each of resolution 32x32. The solution is compressed into latent vectors of size 7. Since the boundary condition is uniform, it is encoded discretely on each subdomain. The boundary condition encoding is a vector of length $8$, such that the first $4$ elements represent an index, $0$ or $1$ indicating Dirichlet or Neumann boundary on the $4$ sides of a subdomain. The next $4$ elements represent the magnitude of BCs on the respective sides. The flux conservation autoencoder has a bottleneck layer of size 35.

### D.1.2 Testing

100 more solutions are generated for arbitrary BCs using Ansys Fluent. The testing set contains completely different boundary conditions than the training set with no overlaps. The convergence tolerance of the CoMLSim solution algorithm is set to $1e^{-8}$ and the solution method is Point Jacobi.

### D.1.3 Comparisons with Ansys Fluent

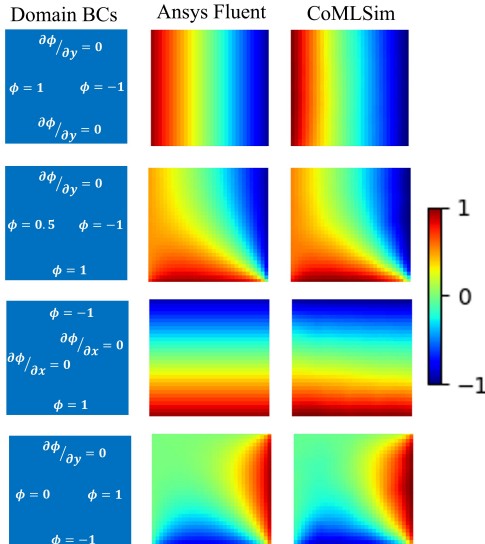

Figure 20: CoMLSim vs Ansys Fluent for Laplace equation over different unseen BCs

The CoMLSim predictions for a selected unseen testing samples are compared with Ansys Fluent in Fig. 20. It may be observed that the contour comparisons agree well with Fluent solutions. Overall, the mean absolute error over 100 testing samples is 0.015.

## D.2 2-D Darcy Equations:

The Darcy equation is defined as follows:

$$-\nabla.(\alpha \nabla \phi(\vec{x}) = f \tag{10}$$

It is subjected to different diffusivity conditions, $\alpha(\vec{x}) = f(\vec{x})$ in a 2-D bounded computational domain between (0, 1). The problem setup as well as the data to train the autoencoders is taken from

[4]. The main objective for including this case was to demonstrate the CoMLSim approach on a public data set and on relatively smaller mesh resolutions.

We use about 400 samples for training the flux conservation autoencoders, and 200 randomly picked samples for testing from the remaining data. The computational domain in this problem is 2D and has a resolution of 241x241. The domain is divided into 241 subdomains of resolution 21x21 each. The solution on each subdomain is encoded into a latent vector of size 11 and the diffusivity is encoded into size 11. The flux conservation network has a bottleneck size of 16. Since, this is a steady state problem, the CoMLSim iterative solution algorithm is initialized with a solution field equal to zero in all test cases. The iterative algorithm convergence tolerance is set to $1e^{-7}$.

The testing error is computed using Eq. 11 shown below:

$$\epsilon = \frac{L_2(Y_{pred} - Y_{true})}{L_2(Y_{true})} \tag{11}$$

The CoMLSim has a testing error of 0.019 averaged over 200 unseen testing samples as compared to a second-order finite difference solver. The CoMLSim error compares well with FNO [4], which has an error of 0.021. Although the testing errors are very similar in this case, the main advantage of the CoMLSim approach is to accurately model local solution features on highly resolved grids and out-of-distribution generalizability as presented in the different use cases in the main paper as well as the supplementary materials.

## E  Computational Speed Analysis:

We observe that the CoMLSim approach is about 40-50x faster as compared to commercial steady-state PDE solvers such as Ansys Fluent for the same mesh resolution in all the experiments presented in this work. Both CoMLSim approach and Ansys Fluent are solved on a single Xeon CPU with single precision. We expect our approach to scale to multiple CPUs as traditional PDE solvers but single CPU comparisons are provided here for benchmarking. Moreover, our algorithm is a python language interpreted code, whereas Ansys Fluent is an optimized, C language pre-compiled code. We expect the C/C++ version of our algorithm to further provide independent speedups (not included in current estimates). In Table 4, we show comparison of simulation time between Ansys Fluent and CoMLSim for all the use cases. The results are averaged over 100 testing cases. As expected, the iterative convergence achieved by the CoMLSim approach is substantially faster than Ansys Fluent.

In comparison to the ML baselines, our approach is expected to be slower because it adopts an iterative inferencing approach. But, our approach focuses on substantially improving the solution accuracy, generalizability, robustness and stability at the cost of computational speed-ups. Although our approach is still faster than traditional PDE solvers, but the ratio of is speed-up is smaller than traditional ML methods.

Table 4: Simulation time (in seconds) comparison with Ansys Fluent

| Experiment | Num. of Elements | CoMLSim | Ansys Fluent |
|---|---|---|---|
| Laplace | 65K | 0.21 | 10 |
| Linear Poissons | 1048K | 2.75 | 130 |
| Coupled non linear Poissons | 1048K | 2.81 | 540 |
| 3D Navier-Stokes flow | 1245K | 35 | 1900 |
| 3D Electronic Chip Cooling | 2097K | 42 | 1600 |

## F  Guidelines for reproducibility

In this section, we provide the necessary details for training the CoMLSim. As emphasized previously, the CoMLSim training corresponds to training several autoencoders for PDE solutions, conditions, such as geometry, boundary conditions and source terms and for flux conservation. In Figure 21, we present a flow chart of the steps that can be followed to train each of these autoencoders. The specific training details and network architectures are described in the previous Section. For a given

set of coupled PDEs and depending of the use case, we start with 100-1000 sample solutions on a computational domain with $n, m, p$ computational elements in spatial directions $x, y, z$, respectively. The solutions are divided into smaller subdomains of resolution, $n/16, m/16, p/16$. Each subdomain can has PDEs solutions and PDE conditions associated with it. The PDE solutions are used as training samples to the PDE solution autoencoder to learn a compressed encoding of all the variables on the subdomain. On the other hand, the autoencoders for PDE conditions can be trained with completely random samples, which may or may not be related to sample solutions. Once the PDE solution and condition autoencoders are trained, neighboring subdomains are grouped together and the solution and PDE condition latent vectors on groups of neighboring subdomains are stacked together. These groups of stacked latent vectors are used for training the flux conservation autoencoder. The trained PDE solution, condition and flux conservation autoencoders combine to form the primary components of the CoMLSim.

The solution algorithm of the CoMLSim has been described in detail in the main body of the paper and may be used for solving for unseen PDE conditions. We are happy to share the source code and the dataset for some experiments if deemed necessary by the reviewers.

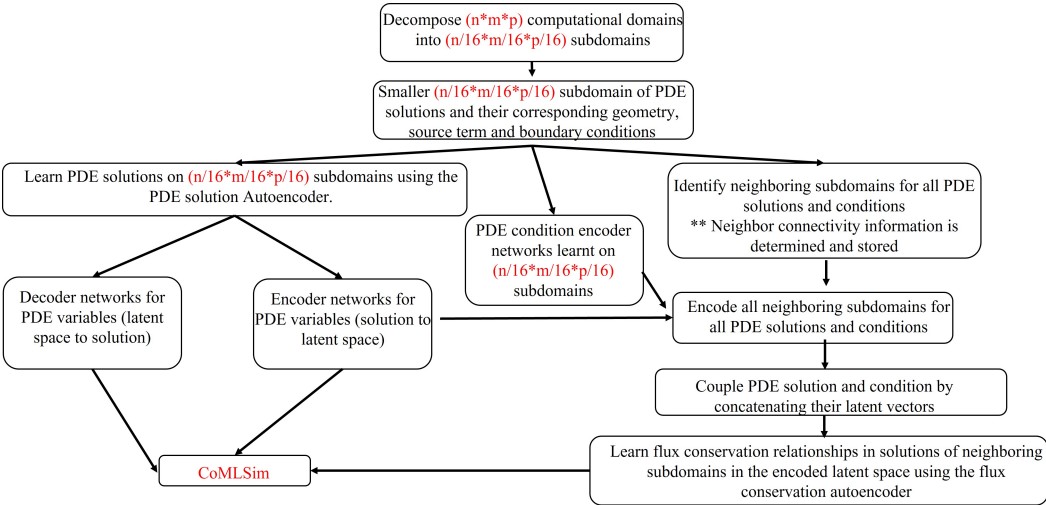

Figure 21: Flow chart for training the CoMLSim approach

## F.1 Step-by-step instructions for Poisson's equation

The Poisson's equation is shown in Eq. 4. The solution to this equation is governed by a source term applied on the computational domain. Here we consider this 2D case to explain the training and inference modes of the CoMLSim approach.

### F.1.1 CoMLSim training

The data in this use case corresponds to 256 training and testing PDE solutions generated for random source terms defined by Eq. 5 using Ansys Fluent on a 1024x1024 resolution grid. This data is preprocessed such that it is divided into subdomains of size 64x64 before being used for training.

The training portion of CoMLSim corresponds to training the solution autoencoder, source term autoencoder and the flux conservation autoencoder. Training solutions and the corresponding conditions on 64x64 subdomains are used for training the solution and condition autoencoders. The Network architecture and training mechanics described in Figure 1 are used to perform this training. Separate 2D CNN autoencoders are used to train solutions and source terms.

The flux conservation networks require further preprocessing of the data before it is trainable. The trained encoders are used to encode the solutions and conditions on each subdomain and its top, bottom, left and right neighbors. These neighborhood encodings are concatenated. For example, if the solution latent size is 16 and the condition latent size is 16, then the input size for the flux conservation autoencoder is (16+16)*5=160. The flux conservation autoencoder is trained to learn

this group of concatenated neighborhood encodings for all the subdomains in the training dataset. For the subdomains on the edges, we simply use zero vectors to pad the missing subdomains. For 3-D cases, we include two additional subdomains, one in front and behind the center subdomain. The network architecture and training mechanics used to perform this training is described in more detail in Figure 2.

The trained autoencoders are used to evaluate the our approach on other unseen source terms.

### F.1.2 CoMLSim inference

During the inference time, we evaluate the CoMLSim approach for a user-specified source term. A schematic of the inference algorithm for this case is shown in Figure. 22.

Similar to training, we decompose the domain into same sized smaller subdomains and encode the solution and source terms on all the subdomains using the pretrained encoders. Since the solution is unknown, a constant or randomly initialized solution is assumed on the domain. In each fixed point iteration, we loop over all the subdomains. On each subdomain, we gather the solution and source term latent vectors of the subdomain and its neighbors. The order in which the neighboring latent vectors are concatenated is consistent with the order used during training. The concatenated neighborhood latent vectors are then passed through the flux conservation autoencoder to obtain a new set of solution and source term encodings for each subdomain and its neighbors. Depending on the algorithm being used, Point Jacobi or Gauss Seidel, we modify the solution encoding of the subdomains while maintaining the source term encoding fixed. The modified solution encodings along with the fixed source term encodings are iteratively passed through the flux conservation autoencoder until the L-2 norm of the change in the solution latent vectors on all subdomains drops below a specified tolerance. The constant source term encodings steer the solution encoding towards an equilibrium. Finally, the converged solution encodings are decoded using the trained solution decoder. The decoded solutions are then stitched together to obtain the full domain solution.

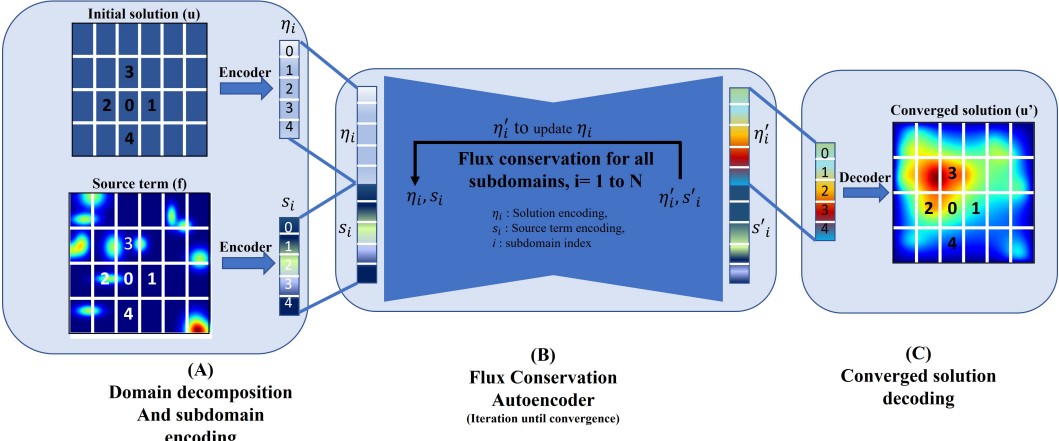

Figure 22: Schematic of CoMLSim inference for Poisson's PDE: The inference process consists of 3 steps. (A) Domain decomposition and encoding: The solution and source term are divided into subdomains. The central subdomain and the neighboring subdomains are encoded individually using the trained autoencoders. $\eta_i$ represents the encoded initial solutions and $s_i$ the encoded subdomains. The solution and source latent vectors on neighboring subdomains are concatenated together. (B) Flux conservation: The concatenated encodings are passed through the flux conservation autoencoder iteratively until convergence. The source term $s_i$, part of the encoding is held constant. Only the solution $\eta_i$, part is updated. In each flux conservation iteration, the algorithm loops over all the subdomains before moving to the next iteration. (C) The converged solution encoding is then extracted from the complete encoding and decoded per subdomain using the solution decoder trained during the training phase.