# OpenReview forum: "A composable machine-learning approach for steady-state simulations on high-resolution grids"
_NeurIPS.cc/2022/Conference — NeurIPS 2022 Accept_

### Official Review · Reviewer_3nTn · 2022-06-29

**Rating:** 6
**Confidence:** 4
**Soundness:** 3 good
**Presentation:** 2 fair
**Contribution:** 3 good

**Summary:**

The paper presents a method to efficiently perform neural simulation of a high-resolution PDE, by dividing the simulation domain into patches, encoding per-patch latents, and learning the simulation on a stencil of neighboring latents.

**Questions:**

I think with a bit of work on clarifying method and results this could be a much stronger paper. If those are addressed sufficiently I will raise my score.

Method:
- The paper talks a lot about using insights from classical PDE solvers, but I couldn't find how those factor into the method. Can you make this clearer? E.g. is there any reason to believe the flux-conservation autoencoder actually preserves flux better than a standard method trained on overall MSE?
- How exactly is the flux-conservation module trained? The simulation output is already at the fix point, so if trained on this data wouldn't it just learn to pass through eta_c and ignore the neighbors?
- Similarly, isn't the distribution shift between train (consistent patches) vs. the first iteration on test (random values with possible discontinuities) a problem?

Experiments/Results:
- How many iterations are needed? How accurate is the model when just run for 1 step?
- Are the baselines also iterated or run one-step?
- How is resolution scaled? CoMLSim could use more patches at test time, but e.g. UNet resolution can't easily be changed without re-training
- Why are the baselines so bad? I certainly have seen much better prediction with UNet/ResNet-like architectures on very similar datasets.
- In general, I can see how this method is more memory efficient and can tackle bigger domains compared to basic UNet, and perhaps better at generalization. But it's very surprising to me why those would have worse accuracy in the in-domain setting; if anything, I would have suspected the reverse, due to UNet's scale hierarchy which is generally very helpful for steady-state PDEs. If it isn't just an undertuned baseline, the reason for this should be properly investigated.


**Limitations:**

There isn't really any discussion on limitations beyond that OOD generalization is hard.

**Strengths And Weaknesses:**

The paper differentiates itself from the common way of learning PDEs by (1) patch-based encoding and (2) iterative application of a local kernel to solve a steady-state problem.
I think the paper's biggest strength is using (1) to solve systems which are otherwise too big to ingest into a NN model; while there are lots of related methods (e.g. GNN-based physics models like MeshGraphNets also operate on local elements to improve generalization), I haven't really seen demonstrations on really big systems like 256^3.
There's quite a few methods which use iterative updates to solve PDEs (e.g. the class of methods surrounding Bai et al.'s Deep Equilibrium Models, or Rubanova et al. Constraint-based graph network simulator), although in a slight different setting. This aspect of the paper however feels very understudied to me (see questions below).
More broadly, while the paper shows plenty of experiments and ablations, I found it very hard to distill concrete findings out of it. It's e.g. unclear what part of the model's performance are due to (1) or (2), and due to vague writing it's unclear how exactly experiments and comparisons were performed. The same is true for the method; it introduces lots of similar-looking symbols which are not properly defined (e.g. how are boundaries and source terms encoded and processed), and the training procedure was very unclear (see questions).
Finally, I'm a bit suspicious of the comparison results. UNets are generally a solid baseline for this kind of problem, but UNet results looked unreasonably bad; this may be a case of a misimplemented or undertuned baseline.

---

> ### Author Response · Authors · 2022-08-02
> **Response to reviewer 3nTn**
>
> We would like to thank the reviewer for the insightful questions and comments. These have been helpful in improving the quality of our paper. All revisions made to the paper and supplementary material are marked in blue.
>
> ### Comments regarding understanding of our method
>
> In section F of supplementary materials, we have added a step-by-step explanation of our approach with an example and an explanation of similarities with traditional PDE solvers in Section A.4. We request the reviewer to review that for more details on our approach.
>
> **“what part of the model's performance are due to (1) or (2), ….”**
> A similar question was raised by Reviewer hAL3. We have conducted additional ablation studies (added to supplementary material section C.1.5) to address this concern. We determine from these experiments that the most important feature of our approach that enhances accuracy, scalability and generalizability is local learning. On the other hand, latent space learning, that is representation of subdomain solutions and conditions with a latent vector, enable faster computations and smaller memory requirements. We request the reviewer to review section C.1.5 of the appendix to get more information. We will add this discussion to the main paper if we are accepted and are able to use an additional page.
>
> **“The paper talks a lot about using insights from classical PDE solvers, but I couldn't find how those factors into the method. Can you make this clearer?”**
>
> As stated in the main paper, there are 3 main similarities between our approach and traditional FVM, FDM based PDE solvers, 1) Domain discretization, 2) Flux conservation and 3) Iterative solution algorithm. Here we provide more details about each one.
>
> 1. **Domain. discretization:** Traditional PDE solvers solve PDEs given in Eq. 1 of our paper by representing solutions variables, u, v, and their spatio-temporal derivatives on a discretized computational domain. The domain is discretized into a finite number of computational elements, using techniques such as Finite Difference Method (FDM), Finite Volume Method (FVM) and Finite Element Method (FEM).
> Similar to traditional PDE solvers, the first step in the CoMLSim is to decompose the computational domain into smaller subdomains. A single subdomain in the CoMLSim is analogous to a computational element in the traditional solver because the CoMLSim predicts PDE solutions directly on local subdomains.
>
> 2. **Flux conservation:** Traditional PDE solvers use numerical approximation schemes are used to compute linear and non-linear components of the PDE. For example, in Eq. 1, if L(u, v) = du/dx + dv/dy, representing the 2-D incompressible continuity equation in fluid flows, the spatial derivatives can be approximated using a second order Euler approximation shown in Eq a and b. These numerical approximations denote flux conservation between neighboring elements. Fluxes represent the flow of information between neighbors and hence, their accurate representation is crucial for information propagation within the domain. Similarly, flux conservation in the CoMLSim happens across neighboring subdomains to ensure local consistency and information propagation. The representation of PDE discretization on subdomains is similar to Equation Eq a, b but the indices i, j represent subdomain indices and the numerical schemes for discretization are now represented by the flux conservation neural network, $\theta$, as shown in Eq c.
>
>     $\left(\frac{\partial u}{\partial x}\right)_{i,j} = $ $ \frac{u(i+1,j) - u(i-1, j)}{2 \delta x}$ -- eq a
>
>     $\left(\frac{\partial v}{\partial y}\right)_{i,j} = $ $ \frac{v(i,j+1) - v(i, j-1)}{2 \delta y}$ -- eq b
>
>     $\left(L(u,v)\right)_{i, j} = \theta (u(i,j), u(i+1,j), u(i-1,j), u(i,j+1), u(i,j-1), v(i,j), v(i+1,j),$
>                                            $v(i-1,j), v(i,j+1), v(i,j-1))$ -- eq c
>
> 3. **Iterative solution procedure:** In traditional solvers, the discretized PDEs represent a system of linear or non-linear equations, where the number of such equations equals the number of computational elements. To solve the PDE solutions, the discretized PDE residual is minimized by enforcing flux conservation iteratively using linear and non-linear equation solvers. Similar to traditional solvers, the discretized PDEs represent a system of linear or non-linear equations, where the number of such equations equals the number of computational subdomains. To solve this system of equations we employ exactly the same techniques that a traditional would use. For example, in this work we have explored 2 linear iterative solution methods, such as Point Jacobi and Gauss Seidel.
>
> This discussion has been added to the Supplementary Materials Section A.4.

---

> > ### Author Response · Authors · 2022-08-02
> > **Continued response to reviewer 3ntn**
> >
> > ### Other comments
> >
> > **"E.g. is there any reason to believe the flux-conservation autoencoder ..."**
> > There is no guarantee that Eq 3 is going to preserve fluxes as Eq a and b would. The idea is to mimic the process of flux conservation using the flux conservation autoencoder. The inner workings of the flux conservation network may be evident in Figure 8 of the supplementary material section C.1.5. In this figure we show the evolution of the Poisson’s equation PDE solution during the iterative inferencing procedure. It may be observed that after first iteration the solution starts evolving from the regions of high magnitude of source terms. As the iterations progress the solution begins to diffuse aligning with the nature of the Poisson’s PDE.
> >
> > **"I haven't really seen demonstrations on really big systems like 256^3"**
> > We have performed another experiment for the RANS external flow use case where we increase the physical domain size to accommodate a mesh resolution of 562x128x128 from the original 304x64x64 and evaluate our approach on an unseen car model. It is important to note that the model is not retrained. The discussion corresponding to this experiment is added in section C.3.4 in supplementary materials.
> >
> > **"How exactly is the flux-conservation module trained? The simulation output ..."** **"Similarly, isn't the distribution shift between train (consistent patches) vs. the first iteration ..."**
> > As described in the Supplementary section A.2, the flux conservation is simply an autoencoder which takes encoding of solutions and conditions on a group of neighboring subdomains as both inputs and outputs. As the reviewer understood, the flux conservation network is in fact trained on converged solutions. At the end of this training, we have a trained network that has learnt only to generate locally consistent PDE solutions (corresponding to a PDE) given a certain condition specified on them. Since this network has only learnt locally consistent solutions, if one were to initialize a group of neighboring subdomains with random noise and iteratively pass it through this network the output of such a procedure would be an equilibrium solution corresponding to some locally consistent PDE solution. The correct PDE solution it converges to depends on the fixed condition specified in this procedure. For these reasons, the distribution shift between training set and first iteration of testing set does not affect the solution convergence.
> >
> > The neighboring eta’s are both inputs and outputs to this network along with the center eta. Hence, the network does not learn to ignore them. The reviewer's suggestion would have been true if the input were eta of neighboring subdomains, but the output was only eta of center subdomain.
> >
> > **"But it's very surprising to me why those would have worse accuracy in the in-domain setting; if anything, I would have suspected the reverse, due to UNet's scale ..."**
> >
> > 1) Traditional ML techniques such as UNet and FCNN are data hungry. In this paper, the number of training samples are equal to 256 for experiments 1 and 2, 150 for experiment 3 and 300 for experiment 4. Also, the experiments considered have large domains and mesh resolutions, for example 1024x1024 for Poisson’s equation, 304x64x64 for external NS flow. The large mesh resolutions significantly increase the parameters space of traditional ml networks and moreover when clubbed with low amounts of data, it results in higher errors for the baselines. With substantial increase in the dataset, both baselines and the CoMLSim approach will continue to improve in accuracy. However, it takes time to generate simulation data and it is preferable to build a model with smaller datasets. Local learning approach eliminates this.
> > 2) Secondly, it should be noted that the large space of input-distributions considered in our experiments results in a wide variation of solution in the domain. The low amount of data to learn this variation exacerbates the issue with accuracy.
> >
> > **"Why are the baselines so bad? I certainly have seen ...”**
> > It should be noted that for in-sample distributions the baselines generally have an acceptable accuracy. These in-sample distribution errors, reported in Table 1, are significantly smaller than the out-of-distribution testing errors. The contours presented in the supplementary materials are for out-of-distribution testing. For example, in figure 4 we significantly increase the number of Gaussians, in figure 5 we completely change the source term distribution, in figure 6 we scale to higher mesh resolution, figure 10 we evaluate on realistic unseen geometries. The baselines trained on high mesh resolutions with low amounts of data do not perform well.

---

> > > ### Author Response · Authors · 2022-08-02
> > > **Continued response**
> > >
> > > **"How many iterations are needed?"**
> > > For the experiments presented in the paper, it requires about 10-200 iterations for convergence. This depends on the complexity of the use case and the method used for solution (Point Jacobi or Gauss Seidel). This information has been added to each use case in the Supplementary Material.
> > >
> > > **"How accurate is the model when just run for 1 step?"**
> > > We have provided a figure in Supplementary Materials Section C.1.5 to show the evolution of the solution from iteration 0 to iteration at convergence. The error after the first iteration is substantially larger than the error observed at convergence.
> > >
> > > **“Are the baselines also iterated or run one-step?”**
> > > The baseline models have been inferred in one-step solution manner. The only model that uses iterative inference is the proposed model.
> > >
> > > **“How is resolution scaled? CoMLSim could use more patches at test time, but e.g. UNet resolution can't easily be changed without re-training"**
> > > The baseline models are not trained again for higher resolutions. The models trained on lower resolution are directly used to infer on higher resolutions. Resolution independence is a well-known feature of the convolution operation. It is done in this way to perform a fair comparison with CoMLSim.

---

> > > > ### Comment · Reviewer_3nTn · 2022-08-05
> > > > **Updated review**
> > > >
> > > > Thank you for the clarifications and for performing additional experiments. This clears up some of my concerns, and strengthens the paper. While some issues remain I will raise my score to 6 (weak accept).
> > > >
> > > > I encourage the authors to incorporate some of the points which came up in review & rebuttal into the main text. In particular,
> > > > 1. A clearer description of the flux autoencoder (how it is trained, why it works) as I think this is currently the point most likely to confuse readers
> > > > 2. Discuss other local (e.g. GNNs) and iterated (e.g. DEQ) approaches in related work
> > > > 3. Make it clear that the baseline comparisons are evaluated in a low-data setting, and that training with more data could produce different outcomes. I think it's totally fine to focus on this regime, but it needs to be made clearer that this is the case.
> > > > 4. The high-res 3d simulations are particularly impressive, as those are usually too big for learned models, I would more prominently feature those in the main text, and maybe include a link to a video.

---

> > > > > ### Author Response · Authors · 2022-08-08
> > > > > **Response**
> > > > >
> > > > > We thank the reviewer for their insightful comments and for raising our score. We appreciate the reviewers suggestions to include in the main text. Due to the 9 page limit we were only able to add points 2 and 3 in the revised main paper at the moment. If the paper gets accepted and we have an additional page, we will definitely add the remaining points to the main paper.

---

### Official Review · Reviewer_1Y4R · 2022-06-30

**Rating:** 7
**Confidence:** 4
**Soundness:** 3 good
**Presentation:** 3 good
**Contribution:** 2 fair

**Summary:**

This paper proposes an algorithm to solve PDEs via an iterative procedure that uses a learned solver for a subdomain. Iterations on multiple of these inferred solutions are performed to obtain a global solution. As such the method represents a hybrid approach between more traditional iterations and learned solvers, with the main goal of achieving solvers for high resolutions without having to re-train models each time. The paper focuses on steady state problems, which limits the scope but at the same time represents a meaningful starting point.


**Questions:**

One aspect that I didnt understand was highlighting that the training is "unsupervised". As explained in the appendix, the training inherently needs many precomputed solutions. This seems misleading to me, unsupervised implies that solutions are not precomputed, or did I overlook something here?


**Limitations:**

I think these are covered quite well via the ablations.

**Strengths And Weaknesses:**

The method is presented in terms of domain subdivisions, with a flux-based coupling for the interfaces between the rectangular subdomains. The iterative solve is performed with an encoded latent space representation for the local solutions. The resulting algorithm is quite straight forward, but the "convergence" condition could be explained a bit more, I believe. This simply seems to be a delta of the latent space vectors, i.e., not a convergence to a true or reference solution, but only one in terms of satisfying the states between neighbors (there's no "residual" or other metric for the overall quality of the solution here).

The first example shown is, unfortunately, directly one of the less convincing ones. The wake behind the "car" and the boundary layer show some clear differences, and the learned solutions seems to have stronger axis aligned artifacts which I guess stem from the piece-wise solutions of the NN. It's good to see a 3D case, though.

The other ablations are interesting, and nicely illustrate the properties of the algorithm. I didn't find the de-noising too convincing, though - this is a classic task for autoencoders, but not a typical issue for numerical solvers.

One aspect that I didnt understand was highlighting that the training is "unsupervised". As explained in the appendix, the training inherently needs many precomputed solutions. This is misleading: unsupervised implies that solutions are not precomputed. Please clarify in the main paper, that the autoencoders are trained with collections of pre-computed sample solutions.

Overall, I think the paper presents an interesting combination of learned and traditional solvers, and combinations of the form proposed here have not been studied a lot. It's good to see how well these piece-wise autoencoders work, and I'm sure the authors can iron out the smaller issues above in a final revision.

---

> ### Author Response · Authors · 2022-08-02
> **Response to reviewer 1Y4R**
>
> We would like to thank the reviewer for reviewing this work. We have addressed the main concerns below. All revisions made to the paper and supplementary material are marked in blue.
>
> ### Reviewer comment regarding unsupervised learning
>
> We agree with the reviewer that our approach is not completely unsupervised since we must generate solution samples to train our autoencoders. Our approach is unsupervised in the sense that we generate training samples, but we don’t use them to learn an explicit relationship between the input and output distribution. Our training consists of simply training autoencoders and the inference algorithm involves solving a constrained fixed-point iteration to converge to a PDE solution. In the constrained fixed-point iteration, the solution converges to a PDE solution starting from initial random noise. Our solution algorithm is never taught this trajectory of solution convergence but discovers that by itself. Hence, we claim that the solution algorithm at inference is unsupervised. We have also added a small discussion in supplementary material section A.5 to clarify this.
>
> However, we agree with the reviewers' concern from a different perspective and are willing to modify this statement in the main paper.
>
> ### Reviewer comment on "The resulting algorithm is quite straight forward, but the "convergence" condition could be explained a bit more, I believe. "
>
> The understanding of the reviewer is correct. Since we don't have access to true solution at the inference stage, we have designed the convergence criterion to be very similar to how traditional steady state solvers measure convergence. In traditional solvers, the convergence is measured by monitoring the change in solution and in our case it is monitored by the change the latent vector of the solution.

---

### Official Review · Reviewer_hAL3 · 2022-07-20

**Rating:** 6
**Confidence:** 3
**Soundness:** 3 good
**Presentation:** 1 poor
**Contribution:** 3 good

**Summary:**

This paper presents a neural network model for solving steady-state PDEs on regular grids. The proposed approach is rooted in the principles of Domain Decomposition, which consists of two major steps: solving PDEs on subdomains in parallel and synchronizing the solutions across the subdomain boundaries, i.e., geometric interfaces partitioning the whole domain. The proposed method makes heavy use of encoders and decoders to solve PDEs in each subdomain. On top of that, the method uses fixed-point iterations to stitch these solutions together.

Roughly speaking, the proposed method can be seen as a deep-learning version of Domain Decomposition solved in a learned latent space.


**Questions:**

I actually like the general idea in this paper, but I think many critical technical contents are described quite vaguely and are difficult to evaluate.

**Problem definition**
1. The proposed approach is tied to a specific spatial discretization scheme (Cartesian grids). While using grids is definitely OK, it should be clearly stated earlier in the paper, e.g., acknowledging it as an assumption in your problem definition.
2. The related work section should also briefly mention papers like GNNs or finite-element methods using triangle or hex meshes so that readers are aware of other spatial discretization schemes.
3. Eqn. (1) is an incomplete definition of PDEs without stating their boundary conditions.
4. The discretization scheme lacks details and misses some subtle points, e.g., how is the smooth geometric surface in the car example handled using your regular grid?
5. Similarly, the concept of “PDE conditions” is also vague and refers to various things in different problems (Sec. A.1.1: “source” in Poisson equation, “geometry” in RANS, physical parameters in other problems, etc.). It is probably better to rename “PDE conditions” so that readers like me don’t mistakenly think it refers to boundary conditions.

**Technical method**
1. Lines 112-114: “Iterative solution algorithms…at convergence.” This statement is specifically for Domain Decomposition, but the context seems to imply all PDE solvers fall into this pattern. Perhaps better to clarify it?
2. Some background knowledge about flux conservation is probably necessary.
3. Overall, I feel Sec. 2.2 and 2.3 lack enough details for people to reproduce the method. The Laplacian example (Eqn. 5) does not help to clarify the method (Eqn. 5 isn’t more informative than Fig 2 or Alg 1). A concrete, step-by-step description of running your approach on a sample problem (Poisson or Laplacian) will be much appreciated.

**Experiments**
While I appreciate the tremendous efforts this paper puts into experiments, I feel many of them are repetitive. Experiments that can tell us what makes combining Domain Decomposition and neural networks a good idea are rare in the paper. One experiment that I really like is Fig. 3C because
- Domain Decomposition is designed for solving large PDE problems due to its superiority in time/spatial complexity;
- Therefore, we have a good reason to hypothesize the proposed method should scale up to resolutions unseen by previous neural network methods;
- Fig. 3C confirms this point with convincing results.

However, other experiments are not very informative to me, although it definitely does not hurt to have them.
- Just like I don’t expect Domain Decomposition to solve PDEs more *accurately* than other traditional methods (CG/multigrid without splitting the domain), I don’t see a strong reason to believe the proposed approach could solve PDEs more accurately than other network models, yet Fig. 3A-3C shows your method achieves smaller MAE than all other baselines across the board. What component in your method makes this happen? Are baselines carefully tuned? I would prefer to see an experiment analyzing it more deeply, e.g., by looking into the differences between your network architecture and that in the baselines.
- Another experiment that I would like to see is to study the influence of latent vector dimensions. As a thought experiment, if we set the latent vector dim = input dim and enforce encoder = identity, does it reduce your method to the standard Domain Decomposition method? By varying the latent vector dim, I would like to understand how it pushes your approach away from the standard Domain Decomposition approach.


**Limitations:**

**Limitations**: Sort of. I think the paper should elaborate more on its limitation (See my comments above), but I can understand it may be difficult due to the page limits and won’t hold it against the paper.

I really appreciate that the authors put “steady-state” in the title to reveal the scope of the PDEs they consider in the first place.

**Potential negative societal impact**: N/A.


**Strengths And Weaknesses:**

Strengths:
1. This paper shows a respectful attempt to combine a classic numerical method with modern deep-learning techniques.
2. The paper presents lots of experiments to evaluate the proposed approach.

Weaknesses:
1. The description of the technical method is vague without rigorous mathematical details. Part of the approach is still mysterious to me.
2. The method and experiments did not reveal much insight from drawing the analogy between Domain Decomposition and neural networks.
I will elaborate on them in the next section.

Some small comments about the figures:
1. It is probably better to place Fig. 2 on page 4.
2. I have found Fig. 19 in the supplemental material helpful. I suggest putting a simplified version of it in the main text if space permits.

---

> ### Author Response · Authors · 2022-08-02
> **Response to Reviewer hAL3**
>
> We would like to thank the reviewer for insightful questions and comments. These have been helpful in improving the quality of our paper. The concerns raised by the reviewer are addressed in our rebuttal below. The comments are inline with the questions raised. All revisions made to the paper and supplementary material are marked in blue.
>
> ### Reviewer comments regarding why our approach out-performs ML baselines.
> The experiments carried out to address these comments are currently added to the Supplementary Material Sections C.1.5 of the revised paper. We will add this discussion to the main paper if we are accepted and are able to use an additional page.
>
> **“What component in your method makes this happen?"**
> Local learning is the main feature that enables our approach to outperform the ML-baselines. We demonstrate the importance of local learning through an experiment. In this experiment we setup the CoMLSim with a single subdomain for the Poisson’s solution use case from the main paper. The solution and source term autoencoders, in this case, are trained on the entire computational domain (1024x1024). Since there are no neighbors, the flux conservation autoencoder is trained on a 1-subdomain stencil. The mean absolute error averaged over 250 test samples is equal to 0.072. Moreover, the computational time required by the single subdomain instance is about 3x more expensive than multi-subdomain instance due to the large latent vector sizes. It is important to note that single subdomain approach cannot scale to larger meshes and the errors are similar to the ML-baselines. This experiment demonstrates the importance of local learning in our approach. More details are provided in Section C.1.5 of supplementary materials under the bullet point "Importance of local learning".
>
> **"Another experiment that I would like to see is to study the influence of latent vector dimensions ..."**
> As suggested by the reviewer, we have added an additional experiment to analyze this. In this experiment, we set the encoder and decoder of solution and condition autoencoders to identity for the Poisson’s solution experiment. By doing this we show that as we reduce to a standard domain decomposition, the accuracy of the approach is maintained but the computational time significantly increases. More details are provided in Section C.1.5 of supplementary materials under the bullet point "Importance of latent space representations".
>
> **"By varying the latent vector dim, I would like to understand how it pushes your approach ..."**
> We have added an additional experiment to analyze this. In this experiment we evaluate the impact of solution and source term latent sizes with accuracy and computational time of our approach. It may be observed from the table that the accuracy decreases as we increase the latent sizes. It is important to note that the result of this experiment cannot be compared to standard domain decomposition. The solution and condition autoencoders in the case of standard domain decomposition were identity and resemble error-free encoding. However, here the autoencoders become overfit for larger latent sizes and substantially diminish the prominence of the latent vector, thereby resulting in bad performance. More details are provided in Section C.1.5 of supplementary materials under the bullet point "Impact of latent vector size".
>
> | Solution encoding size | Mean Absolute Error | # of Iterations |
> |:---:|:---:|:---:|
> | 7 | 0.047 | ~75 |
> | 11 | 0.017 | ~100 |
> | 32 | 0.029 | ~250 |
> | 64 | 0.1 | ~600 |
> | 128 | 0.21 | ~700 |
> | 256 | 0.29 | ~800 |
>
> ### Reviewer comments on ML-baselines
>
> **“Are baselines carefully tuned? I would prefer to see an experiment analyzing it more ..."**
> Yes, the baselines are carefully tuned. In Figure 12 of supplementary material, we provide the ML baseline training curves for one of the experiments to show this. Also, Table 1 in the main paper shows that for most experiments the baselines have an adequate performance for in-distribution testing. Most traditional ML methods are data hungry. In all the experiments considered in the paper, we use 200-300 training samples to solve PDEs on large domains with resolutions of 1024x1024, 304x64x64 etc with widely varying and high-dimensional input and output distributions. These reasons contribute to why our approach outperforms ML baselines. Data generation is costly in numerical simulation. So, the preference is to have lesser number samples and focus more on local learning.
>
> Our training and evaluation methodologies are different from traditional ML techniques, so it is difficult to compare the network architectures with the baselines.
>
> ### Reviewer comment on step-by-step instructions of our approach with an example
> We have added this to the supplementary materials Section F.

---

> > ### Author Response · Authors · 2022-08-02
> > **Continued response to Reviewer hAL3**
> >
> > ### Reviewer comment on repetitive experiments
> > In the main paper experiments, we have a good mix of pure research and engineering problems with varying levels of non-linear complexity, different input distributions, PDEs, solution variable coupling, spatial dimension etc.
> >
> > For example,
> >
> > 1. In experiment 1, we solve the 2-D Poisson’s equation for a high-dimensional input distribution of source terms, specified in Eq. 8.
> >
> > 2. In experiment 2, for a similar source term distribution we add non-linear complexity to the problem by including an additional PDE, where the solution variables are tightly coupled.
> >
> > 3. In experiment 3, we consider a realistic case of 3-D external flow over arbitrary objects. Here the input distribution is changed to geometry. The PDEs are elliptic/hyperbolic in nature and the system of PDE consists of 4 solution variables (3 components of velocity and pressure). The non-linearity in this case results explicitly from geometry and implicitly from the convective (for example d (density x velocity x velocity) / dx) terms in the PDE.
> >
> > 4. In experiment 4, we show a realistic industrial process modeling the complicated physics of natural convection, which results from a coupling between velocity and temperature.
> >
> > 5. In experiments in the Supplementary Materials section D.1 and D.2 show extension to different input distributions such as diffusivity and boundary conditions.
> >
> > We have added this discussion to Appendix C of the supplementary materials.
> >
> > ### Minor comments
> >
> > **"background knowledge about flux conservation is probably necessary”**
> > This discussion has been added to supplementary material section F and section A.4
> >
> > **“Lines 112-114: “Iterative solution algorithms…at convergence ...”**
> > We have now clarified in the paper that we are referring to PDE solvers based on FEM, FVM and FDM methods.
> >
> > **“Eqn. (1) is an incomplete definition of PDEs without stating their boundary conditions.”**
> > This has been corrected in the paper.
> >
> > **“The discretization scheme lacks details and misses some subtle points ...”**
> > In the supplementary material section C, for each experiment, we have specific details regarding the discretization used for training the model.
> >
> > Regarding the geometry representation, the representation used is based on the Signed Distance Field (SDF). The section 2.3 in the main paper mentions the usage of SDF for geometry representation. Also, the section A1.1 in the supplementary materials explains SDF in more details.
> >
> > **“The related work section should also briefly mention papers like GNNs ...”**
> > The related works section is modified to reflect this change.
> >
> > **“The proposed approach is tied to a specific spatial discretization scheme (Cartesian grids). While using grids ...”**
> > This has been identified as an extension to the current approach in Section 5 of the main paper. Although we use Cartesian grids in this paper, we believe it can be extended to unstructured grids by modifying the solution and condition autoencoders to encode and decode unstructured subdomain.  Also, we have added this clarification in the introduction of the paper.

---

> > > ### Author Response · Authors · 2022-08-08
> > > **Thank you and following up**
> > >
> > > Thanks for acknowledging our rebuttal. Your feedback was useful. Please let us know if you have any additional thoughts or comments that we can address to put our paper in a better standing.

---

> > > > ### Comment · Reviewer_hAL3 · 2022-08-09
> > > > **Thank you for the revision**
> > > >
> > > > Thank you very much for the careful revision. Many of my questions have been addressed, and I will raise my score to 6.
> > > >
> > > > I have a few very minor comments (not critical to my score):
> > > > - Adjust the text/figure 5 layout to remove the space in lines 325-331;
> > > > - Sec. C.3.4: I suggest that the opening paragraph states explicitly that the more complex geometry is voxelized on a background grid (i.e., it is not a smooth surface that can cut through voxels).

---

### Comment · Area_Chair_S5yY · 2022-08-10
**Rebuttal Acknowledgement**

Dear Reviewers,

We are entering the discussion phase, where the authors will be not involved in the discussion.

I would like to request you to confirm that you have already read the rebuttal from the authors.

Best

AC

---

### Meta-Review · Area_Chair_S5yY · 2022-08-26

**Recommendation:** Accept
**Confidence:** Less certain

**Metareview:**

This papers proposed a new method to predict accurate and generalizable PDE solutions on high-resolution grids. All reviewers found the paper is interesting and are positive about the paper. Please address the reaming concerns in the next version.

**Award:**

No

---

### Decision · Program_Chairs · 2022-09-14

Accept